# Low elevation of Svalbard glaciers drives high mass loss variability

Brice Noël [1✉], C. L. Jakobs [1], W. J. J. van Pelt [2], S. Lhermitte [3], B. Wouters [1,3], J. Kohler[4], J. O. Hagen[5], B. Luks [6], C. H. Reijmer [1], W. J. van de Berg[1] & M. R. van den Broeke [1]

Compared to other Arctic ice masses, Svalbard glaciers are low-elevated with flat interior accumulation areas, resulting in a marked peak in their current hypsometry (area-elevation distribution) at ~450 m above sea level. Since summer melt consistently exceeds winter snowfall, these low-lying glaciers can only survive by refreezing a considerable fraction of surface melt and rain in the porous firn layer covering their accumulation zones. We use a high-resolution climate model to show that modest atmospheric warming in the mid-1980s forced the firn zone to retreat upward by ~100 m to coincide with the hypsometry peak. This led to a rapid areal reduction of firn cover available for refreezing, and strongly increased runoff from dark, bare ice areas, amplifying mass loss from all elevations. As the firn line fluctuates around the hypsometry peak in the current climate, Svalbard glaciers will continue to lose mass and show high sensitivity to temperature perturbations.

[1] Institute for Marine and Atmospheric research Utrecht, Utrecht University, 3584 CC Utrecht, Netherlands. [2] Department of Earth Sciences, Uppsala University, SE 75236 Uppsala, Sweden. [3] Department of Geoscience & Remote Sensing, Delft University of Technology, 2600 AA Delft, Netherlands. [4] Norwegian Polar Institute, N-9296 Tromsø, Norway. [5] Department of Geosciences, University of Oslo, 0371 Oslo, Norway. [6] Institute of Geophysics, Polish Academy of Sciences, 01-452 Warsaw, Poland. ✉email: b.p.y.noel@uu.nl

Glaciers and ice caps in the Svalbard archipelago (Fig. 1a) cover an area of ~34,000 km², representing about 6% of the world's glacier area outside the Greenland and Antarctic ice sheets[1]; they contain 7740 ± 1940 km³ (or Gigaton; Gt) of ice, sufficient to raise global sea level by 1.7 ± 0.5 cm if totally melted[2]. As a result of Arctic Amplification[3], in which Arctic warming over the last two decades was twice the global average[4], and being situated at the edge of retreating Arctic sea ice, Svalbard ice caps experience among the fastest warming on Earth. Compared to other Arctic ice caps, Svalbard glaciers have relatively low elevations (Fig. 1b). The highest elevation on Svalbard is ~1700 m above sea level (a.s.l.), but the glacier hypsometry (area-elevation distribution) peaks at ~450 m a.s.l. compared to 800–1400 m a.s.l. for ice caps in Greenland, Arctic Canada and Iceland (Fig. 1b). About 60% of the total glacier area of Svalbard is located below that hypsometry peak. Moreover, Svalbard ice caps have relatively flat interior accumulation zones leading to a more pronounced peak compared to other Arctic ice masses (Fig. 1b).

Combined in situ and remote-sensing measurements show that Svalbard land ice has been losing mass at strongly fluctuating rates since the early 2000s[1,5–12]. According to gravity recovery and climate experiment (GRACE) data, mass loss virtually stopped in 2005–2012, between two periods of sustained mass loss (2002–2004 and 2013–2016)[10]. Glacial mass balance (MB) expresses the difference between the surface mass balance (SMB) and solid ice discharge (D). Glacial mass loss can thus originate from increased D from accelerating marine-terminating glaciers[13], and/or a decrease in SMB, the difference between mass accumulation from snowfall and ablation mainly from meltwater runoff. Surge-type glaciers strongly impact D and are widespread in Svalbard[14], with more than 700 glaciers that likely surged in the past[15]. Although surge events can strongly influence mass loss locally[16], these events are poorly understood and are only documented for a few glaciers[17–19]. Here we use a Svalbard-wide solid ice discharge estimate for the period 2000–2006[13], complemented by an increase in D after the surge of a major Austfonna (AF) glacier in 2012–2013[20].

While ice discharge can be derived from remote sensing, surface processes driving the SMB of Svalbard glaciers remain poorly constrained. Regional climate models can, in principle, represent the SMB of Svalbard glaciers[21,22], including internal accumulation of rain and meltwater in firn through refreezing (see "Methods" section). However, these models currently operate at relatively coarse spatial resolutions, typically 5–20 km, and do not resolve the narrow marginal ablation zones and outlet glaciers[23,24]. In previous studies, regional climate model outputs were refined to higher spatial resolution, e.g. 250 m to 1 km, using positive degree day[25] or energy balance models[26,27] to show that Svalbard recently lost mass following an increase in summer ablation (Supplementary Table 1). Similar conclusions were drawn by upscaling in situ SMB measurements to all Svalbard land ice[12], but little remains known about the temporal and spatial variabilities of the surface mass loss.

Statistical downscaling to (sub-)km horizontal resolution[28] is a powerful tool to realistically represent the steep SMB gradients in the topographically complex terrain that characterises the Svalbard archipelago. Here we present and evaluate a new, high-resolution daily SMB data set for Svalbard covering the period 1958–2018 (Fig. 1a). SMB components are statistically downscaled from the output of the regional atmospheric climate model (RACMO2.3) at 11 km resolution[29] to a glacier mask and digital elevation model (DEM) on a 500 m horizontal grid (Supplementary Fig. 1). The method primarily corrects daily melt and

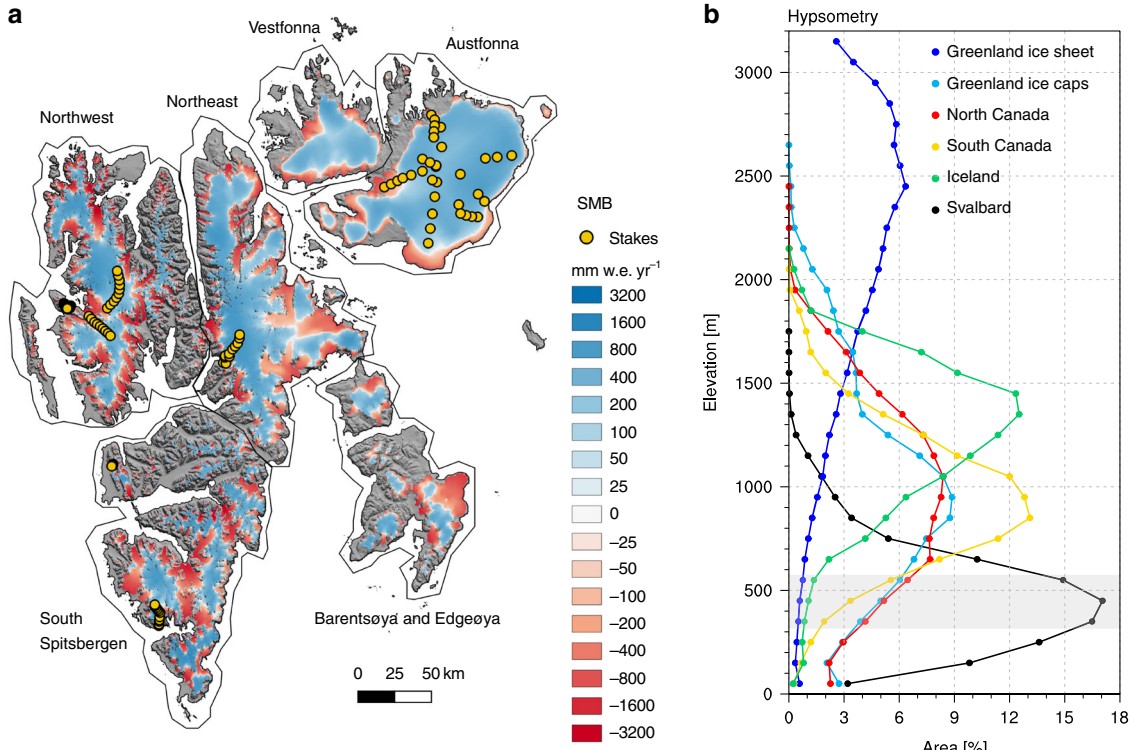

**Fig. 1 Svalbard surface mass balance and hypsometry. a** Modelled surface mass balance (SMB) statistically downscaled to 500 m spatial resolution, averaged for the period 1958–2018. Orange dots locate the 101 stakes used for model evaluation (Supplementary Fig. 2a). The sectors of Svalbard evaluated in Supplementary Fig. 2b are also outlined. **b** Hypsometry of six Arctic ice masses: Svalbard (S0 Terreng DEM), Iceland (Arctic DEM), North and South Canadian Arctic Archipelago (Canadian DEM)[24], Greenland ice sheet (GIMP DEM)[41], Greenland peripheral glaciers and ice caps (GIMP DEM)[23]. The x-axis shows the glacier area in each 100 m elevation band as a fraction of the total ice area of that region (%).

runoff for elevation biases on the relatively coarse RACMO2.3 model grid using elevation gradients, and for underestimated ice albedo using remote-sensing measurements[28] (see "Methods" section). The new product includes all individual SMB components (snowfall, rainfall, sublimation, melt, refreezing, and runoff) required to identify the drivers of the recent surface mass loss and its variability. Combined with discharge estimates[13,20], our high-resolution SMB product enables us to estimate the spatially and temporally varying MB of Svalbard glaciers over the last six decades, including the high mass loss variability starting in the mid-1980s. We show that a modest atmospheric warming of 0.5 °C in the mid-1980s was sufficient to raise the firn line to the hypsometry peak at ~450 m a.s.l., exposing large parts of the accumulation area to increased melt. The subsequent loss of refreezing capacity, i.e. the fraction of rain and meltwater retained or refrozen in firn (see "Methods" section), implies that Svalbard ice caps can no longer be sustained when the current climate persists or further warming occurs.

## Results

**Model evaluation.** The SMB product is evaluated using 1611 local (in situ) annual balance measurements from 101 sites (Fig. 1a) collected in the ablation and accumulation zones of Svalbard glaciers over the period 1967–2015 (see "Methods" section; Supplementary Fig. 2a). Good agreement with the SMB product is found ($R^2 = 0.63$), with a small positive bias of 5 mm w.e. yr$^{-1}$ (water equivalent). Note that significant deviations (RMSE) of up to 440 mm w.e. yr$^{-1}$ remain locally (Supplementary Fig. 2a). Unlike the downscaled SMB product, stake measurements in the accumulation zone do not include internal accumulation from the refreezing of melt and rain (see "Methods" section). Ignoring internal accumulation when comparing the model to stake measurements located in the accumulation zone leads to a small RMSE increase of ~50 mm w.e. yr$^{-1}$. We estimate an uncertainty in total Svalbard SMB of 1.6 Gt yr$^{-1}$ (~25%) for the period 1958–2018 (see "Methods" section). Using data from the moderate resolution imaging spectroradiometer (MODIS) satellite over 2000–2018, we also evaluate the modelled bare ice area, i.e. the part of the ablation zone where bare ice is exposed after the seasonal snow has melted (Supplementary Fig. 2b). To that end, we divide Svalbard into six sectors (Fig. 1a) namely Northwest (NW), Northeast (NE), Vestfonna (VF), AF, Barentsøya and Edgeøya (BE), and South Spitsbergen (SS). With 93% of the variance explained and an average negative bias of

90 km$^2$, modelled and observed bare ice area compare very well (Supplementary Fig. 2b).

We assume that solid ice discharge estimate for 2000–2006 ($D = 6.8 \pm 1.8$ Gt yr$^{-1}$)[13] is valid for the whole study period (1958–2018). In line with Dunse et al. (2015)[20], we increase solid ice discharge by $4.2 \pm 1.6$ Gt yr$^{-1}$ from 2012 onwards, following the surge of a major AF outlet glacier. Combining this with the downscaled SMB product, we reconstruct the mass change of Svalbard glaciers over the last six decades (Fig. 2). The modelled mass change is obtained by integrating both SMB and $D$ in time starting from zero in 1958. Our reconstruction agrees very well with remote-sensing records from GRACE (2002–2016)[10] and ICESat/CryoSat-2 altimetry (2003–2018) with $R^2 = 0.93$ and 0.98, respectively (Supplementary Fig. 2c). Not only the recent mass trends but also the seasonal and interannual variabilities are accurately reproduced. Supplementary Table 1 compares our results to other mass change estimates derived from geodetic techniques[1,11], GRACE[5–8,10], SMB models including a positive degree day[25], two energy balance models[26,27], two regional climate models[21,22], and in situ measurements[12].

**Recent mass loss onset.** Our reconstruction shows that Svalbard glaciers remained in approximate balance (SMB ≈ D) until the mid-1980s (Fig. 2), i.e. the surface mass gain compensates the dynamic mass loss from calving[13]. Net mass loss starts around 1985, primarily due to a persistent SMB decrease, reinforced from 2012 onwards by enhanced ice discharge[20], but with a mass loss pause between 2005 and 2012. Our reconstruction suggests that Svalbard has lost ~350 Gt of ice since 1985, contributing ~1 mm to global sea level rise (Fig. 2). Both remote-sensing data and our reconstruction show that Svalbard glaciers have experienced mass loss since the mid-1980s, including the pause between 2005 and 2012. Understanding the drivers of the pronounced post-1985 mass loss variability requires investigating spatial and temporal fluctuations in individual SMB components.

**Ablation zone expansion and firn line retreat.** Figure 3a shows time series of individual SMB components covering the period 1958–2018. The ice caps of Svalbard experience average summer melt (1958–1984 average of 28.7 Gt yr$^{-1}$, Supplementary Table 2) that exceeds annual total precipitation (23.0 Gt yr$^{-1}$ including rain and snow) by 25%. This proves that retention of surface meltwater in the firn through refreezing is crucial to sustain these ice caps. The refreezing capacity is defined as the fraction of

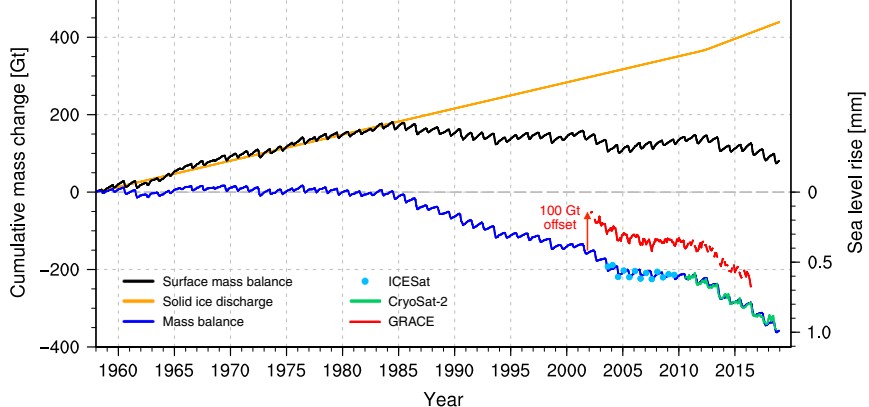

**Fig. 2 Cumulative mass change of Svalbard glaciers and contribution to sea level rise.** Time series of monthly cumulative modelled SMB, measured cumulative solid ice discharge ($D$)[11,12] and reconstructed cumulative mass balance (MB = SMB−$D$) for the period 1958–2018. Observed mass change derived from GRACE (2002–2016), ICESat (2003–2009) and CryoSat-2 (2010–2018) are also shown. For clarity, GRACE data are shown with a positive offset of 100 Gt. The right $y$-axis translates Svalbard cumulative mass balance into global sea level rise equivalent. Supplementary Fig. 2c zooms in on the satellite period (2003–2018).

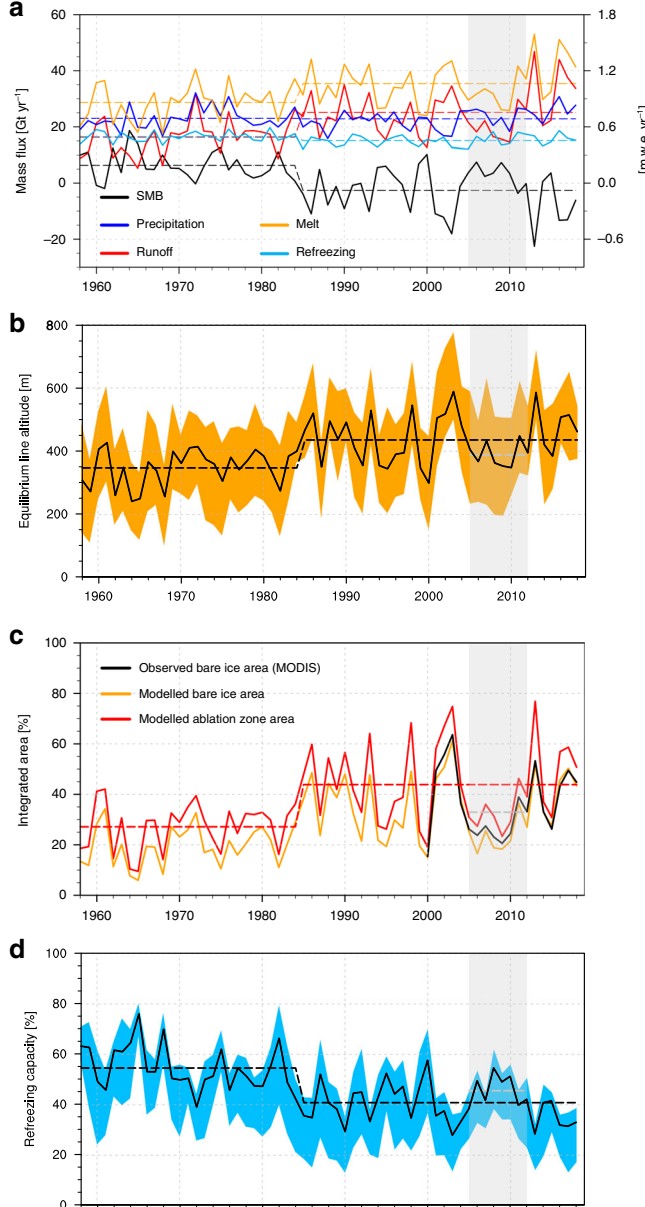

**Fig. 3 Ablation zone expansion and reduced refreezing capacity. a** Time series of annual SMB and components including surface melt, runoff, total precipitation, and refreezing for the period 1958–2018. **b** Time series of annual ELA for the whole of Svalbard (black) and individual sectors (Fig. 1a, orange band). **c** Time series showing the modelled ablation zone area, the modelled and observed (MODIS) bare ice area as a fraction of the total Svalbard land ice area (%). **d** Time series of annual refreezing capacity for the whole of Svalbard (black) and individual sectors (cyan band). Dashed lines show averages for the periods 1958–1984 and 1985–2018. The grey shade highlights the period 2005–2012 when Svalbard SMB temporarily returned to the pre-1985 SMB conditions. Dashed grey lines represent the 2005–2012 mean conditions.

liquid water (melt and rain) that is retained in the firn. Before 1985, the refreezing capacity was 54%, reducing meltwater runoff (16.3 Gt yr$^{-1}$) and resulting in a positive SMB (6.3 ± 1.6 Gt yr$^{-1}$; Fig. 3a). This surface mass gain was almost exactly offset by solid ice discharge (6.8 ± 1.8 Gt yr$^{-1}$)[13].

Following a modest atmospheric warming (+0.5 °C; 1985–2018 minus 1958–1984), the average equilibrium line altitude

(ELA; local SMB = 0) moved upwards by ~100 m, from ~350 to ~450 m a.s.l. (Fig. 3b). The orange band in Fig. 3b spans the six regional ELA values, the change ranging from +80 m in SS to +130 m in the NE sectors (Supplementary Tables 2 and 3). The ELA increase caused a rapid retreat of the firn line, as shown by the post-1985 growth of the bare ice zone (+75%; Fig. 3c) in good agreement with MODIS records (see "Methods" section). As a result, the ablation zone expanded from 27% to 44% of the total glacier area (Fig. 3c). While total precipitation did not significantly change after 1985 (−1%), surface melt increased by 24%, exceeding accumulation by 58%, while the refreezing capacity declined from 54% (1958–1984) to 41% (1985–2018; Fig. 3d). The blue band in Fig. 3d spans the six individual regions that underwent a simultaneous and similar decline in refreezing capacity, ranging from 22% in NW to 36% in BE sectors, respectively (Supplementary Tables 2 and 3). Consequently, SMB became predominantly negative (−2.6 ± 1.6 Gt yr$^{-1}$), initiating the post-1985 mass loss of Svalbard glaciers. We conclude that all regions in Svalbard experienced rapid ablation zone expansion and reduced firn refreezing capacity, resulting in strongly increased meltwater runoff (+55%), driving the post-1985 glacial mass loss (MB = −10.2 ± 3.4 Gt yr$^{-1}$; Supplementary Table 3).

**Discussion.** Compared to other Arctic ice masses[23,24], Svalbard glaciers have a low elevation and are relatively flat with a marked hypsometry peak at ~450 m a.s.l. (Fig. 1b). Before 1985, the ELA was at 350 ± 60 m a.s.l., well below the hypsometry peak (Figs. 1, 3b and Supplementary Fig. 3a). In this period, 70% of the total glacier area was covered with extensive firn zones, in which most meltwater and rain were refrozen. This kept the SMB positive, as runoff remained smaller than snow accumulation (Fig. 3a). Following a modest atmospheric warming after 1985, the ELA moved upward by ~100 m to 440 ± 80 m a.s.l. (Fig. 3b and Supplementary Fig. 3b), nearly coinciding with the hypsometry peak (Supplementary Fig. 3d). This rapidly expanded the ablation zone, exposing large areas to increased melt. The subsequent firn line retreat strongly reduced the fraction of melt that refreezes above the pre-1985 ELA (Fig. 3d), enhancing runoff 75% faster than melt (+8.9 vs. +6.7 Gt yr$^{-1}$). Supplementary Fig. 4a shows the ELA change across Svalbard as a result of the post-1985 warming (R = 0.82; Fig. 4a). The ablation zone extent increases non-linearly with the upward migration of the ELA (Fig. 4b), reflecting the proximity of the hypsometry peak (Fig. 3b, c). The size of the ablation zone in turn governs meltwater production (Fig. 4c), since most of the melt is produced over low-lying marginal glaciers exposing dark bare ice (Supplementary Fig. 4b). In the absence of refreezing, the low albedo of exposed ice increases melt through enhanced absorption of incoming solar radiation, in turn driving the runoff increase. Most remarkably, increased melt triggers a pronounced non-linear decrease in refreezing capacity (Fig. 4d), as (i) the firn line retreat strongly reduces the firn area hence limiting meltwater retention, and (ii) meltwater fills the pore space of the remaining firn through refreezing. These mechanisms could likely be reinforced by increased rainfall episodes in a warmer climate, further reducing firn refreezing capacity[30].

Regionally, the upward migration of the ELA is largest in the northernmost sectors, e.g. NE (+130 m) and AF (+120 m), compared to southern sectors with an average of +85 m (Supplementary Tables 2 and 3). As a result, the ablation zone also grew fastest in the north, e.g. NE (+73%), VF (+91%), and notably AF (+137%; Supplementary Fig. 4a) compared to southern sectors (+48% on average; Supplementary Tables 2 and 3). For the northern sectors, this resulted in a 66–71% runoff increase after 1985, i.e. well above the Svalbard average (+55%;

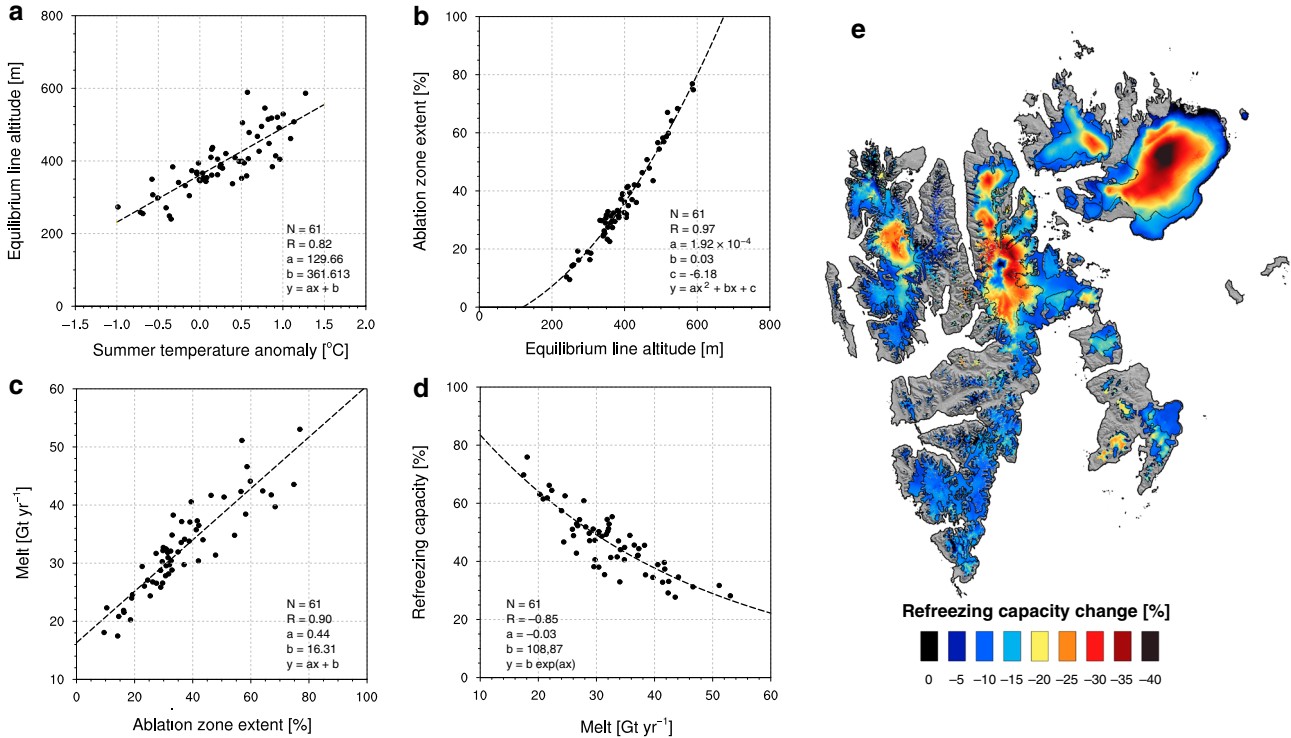

**Fig. 4 Sensitivity of Svalbard refreezing capacity to atmospheric warming.** Scatter plots showing Svalbard-wide correlations between **a** June–July–August 2 m air temperature anomaly (1985–2018 minus 1958–1984) and ELA. **b** ELA and ablation zone area, **c** ablation zone area and surface melt, and **d** melt and firn refreezing capacity. Statistics include number of records (*N*), correlation (*R*), and fitting parameters (*a–c*). **e** Post-1985 change in refreezing capacity (%; 1985–2018 minus 1958–1984). ELA for the period 1985–2018 is also shown as a black line.

Supplementary Tables 2 and 3). These three northernmost sectors exhibit a stronger response to atmospheric warming because of a pronounced decline in refreezing capacity across their accumulation zones (-40% locally; Fig. 4d, e), increasing runoff at all elevations (Supplementary Fig. 4b). These results are in line with the study of Van Pelt et al. (2019) (see their Fig. 9d)[27]. Since it has the largest accumulation zone, the strongest sensitivity to atmospheric warming is found for AF ice cap (AF sector), containing a third (~2500 km³)[16] of the total ice volume in the archipelago. In contrast, for regions with smaller accumulation zones (NW and SS) or that had already lost most of their refreezing capacity before 1985 (BE; Supplementary Table 2), the runoff increase is restricted to the margins (Supplementary Fig. 4b), and primarily driven by ablation zone expansion rather than loss of refreezing capacity (Fig. 4c).

The fact that the ELA now fluctuates around the hypsometry maximum makes Svalbard glaciers highly sensitive to changes in atmospheric temperature. During warm summers, the ablation zone now covers more than half of the surface area of most ice caps (Fig. 3c). In the warm summer of 2013, the ablation zone even covered 77% of the land ice area (Fig. 5b), almost twice the post-1985 average (44%; Supplementary Table 3). This pronounced expansion stems from the fact that in 2013 the ELA moved to 590 m a.s.l., i.e. above the hypsometry peak (Supplementary Fig. 3d). Consequently, the refreezing capacity dropped to 28% (2013), more than doubling runoff compared to previous years (47 Gt yr⁻¹; Fig. 3a). We conclude that the post-1985 decline in refreezing capacity will persist under continued warming: a temporary return to pre-1985 SMB values in the period 2005–2012 (Figs. 3a and 5a) did not lead to the recovery of the refreezing capacity (Fig. 3d). At the current mass loss rate (19.4 ± 3.4 Gt yr⁻¹ for 2013–2018), Svalbard glaciers would completely melt within the next 400 years.

## Methods

**Regional climate model and statistical downscaling**. We use the outputs of RACMO2.3[29] as input to the statistical downscaling procedure[28]. RACMO2.3 is run at 11 km spatial resolution for the period 1958–2018. The model incorporates the dynamical core of the high-resolution limited area model (HIRLAM)[31] and the physics of the European Centre for Medium-Range Weather Forecasts-Integrated Forecast (ECMWF-IFS cycle CY33r1)[32]. RACMO2.3 includes a multi-layer snow module simulating melt, water percolation, retention and refreezing in firn, as well as runoff[33]. The model accounts for dry snow densification[34], drifting snow erosion and sublimation[35], and explicitly simulates snow albedo[36]. In this study, we refer to 'SMB' as both the local (kg m⁻²yr⁻¹) and spatially integrated (Gt yr⁻¹) sum of:

$$\mathrm{SMB} = \mathrm{PR} - \mathrm{RU} - \mathrm{SU} - \mathrm{ER} \qquad (1)$$

where PR represents total precipitation including snowfall (SF) and rainfall (RA), RU meltwater runoff, SU total sublimation and ER the erosion from drifting snow. Liquid water from rain and melt (ME) that is not retained or refrozen in firn (RF) contributes to runoff:

$$\mathrm{RU} = \mathrm{ME} + \mathrm{RA} - \mathrm{RF} \qquad (2)$$

Note that in Cogley et al. (2011)[37], the local quantity that includes 'internal accumulation' from refreezing and retention (RF) is referred to as 'climatic mass balance'. Firn refreezing capacity (RFcap), i.e. the fraction of rain and meltwater effectively retained or refrozen, is estimated as

$$\mathrm{RFcap} = \frac{\mathrm{RF}}{\mathrm{ME} + \mathrm{RA}} \qquad (3)$$

RACMO2.3 is forced by ERA-40 (1958–1978)[38] and ERA-Interim (1979–2018)[39] reanalyses on a 6-hourly basis within a 24 grid-cell wide relaxation zone at the 40 vertical atmospheric levels. The model also includes 40 active snow layers that are initialised in September 1957 using vertical temperature and density profiles derived from the Institute for Marine and Atmospheric research Utrecht-Firn Densification Model (IMAU-FDM)[34]. In RACMO2.3 Svalbard firn can be 30–40 m deep locally. Bare ice albedo is prescribed from a down-sampled version of the 500 m MODIS albedo 16-day product (MCD43A3) as the 5% lowest surface albedo records for the period 2000–2015, minimised at 0.30 for dark bare ice and maximised at 0.55 for bright ice beneath perennial firn.

To resolve narrow ablation zones and small glaciers of Svalbard, the outputs of RACMO2.3 are statistically downscaled to a 500 m ice mask derived from the Randolph Glacier Inventory (RGI)[40] version 6.0 and the 20 m spatial resolution S0 Terreng DEM of Svalbard (Norwegian Polar Institute) down-sampled onto a 500 m grid (Supplementary Fig. 1). In brief, the downscaling procedure corrects

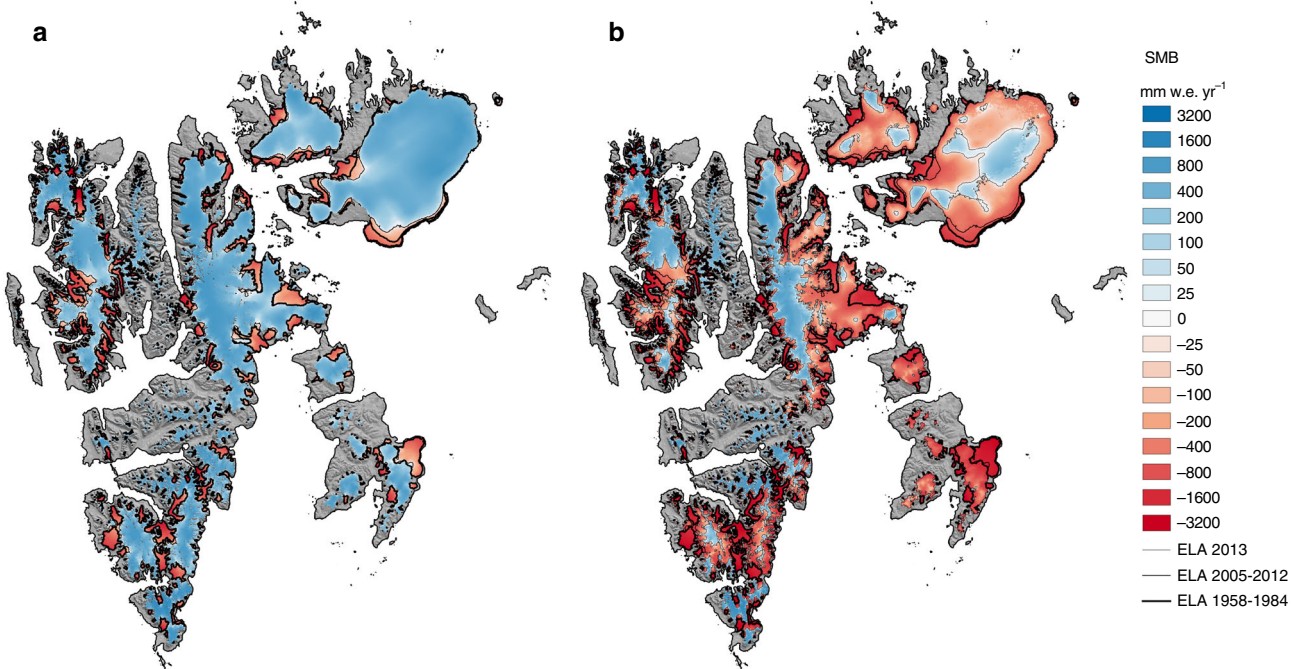

**Fig. 5 Ablation zone expansion in summer 2013. a** SMB average for the period 2005–2012, with SMB conditions similar to 1958–1984. **b** SMB for year 2013 highlighting how fast the ablation zone expands when the ELA migrates well above the hypsometry maximum (~450 m a.s.l.). From the thickest to the thinnest, black lines outline the ELA for periods 1958–1984, 1985–2018 (**a** and **b**) and year 2013 (**b** only).

individual SMB components (except for total precipitation), i.e. primarily meltwater production and runoff, for elevation and ice albedo biases on the relatively coarse model grid at 11 km resolution. These corrections reconstruct individual SMB components on the 500 m topography using daily specific gradients estimated at 11 km, and minimise the remaining runoff underestimation using a down-sampled 500 m MODIS 16-day ice albedo product averaged for 2000–2015 [https://doi.org/10.5067/MODIS/MCD43A3.006]. Total precipitation, including SF and RA, is bilinearly interpolated from the 11 km onto the 500 m grid without additional corrections. The statistical downscaling technique is further described in Noël et al. (2016)[28].

**Product uncertainty**. The SMB uncertainty ($\sigma$) is estimated at an average of 1.6 Gt yr$^{-1}$ for the period 1958–2018. The uncertainty is obtained by integrating the conservative 10% and 20% SMB uncertainty in RACMO2.3[41] over the accumulation ($A_{\text{accum.}}$ = 21,100 km$^2$) and ablation zones ($A_{\text{abla.}}$ = 11,650 km$^2$) of Svalbard, respectively. A similar uncertainty is estimated for individual sectors (Supplementary Tables 2 and 3) following:

$$\sigma = \sqrt{(0.1 \times A_{\text{accum.}})^2 + (0.2 \times A_{\text{abla.}})^2} \qquad (4)$$

**Modelled ELA**. To estimate the modelled ELA (local SMB = 0), we used the down-sampled S0 Terreng DEM of Svalbard at 500 m to average the surface elevation of grid cells showing an annual cumulative SMB ranging from −50 to 50 mm w.e. for each specific year. The procedure was conducted separately for the six sectors and the whole of Svalbard over the periods 1958–1984 (Supplementary Table 2) and 1985–2018 (Supplementary Table 3). We estimated the associated uncertainty as one standard deviation of the annual ELA for the two periods and for each individual sectors. We repeated the procedure using various thresholds ranging from 5 to 100 mm w.e. and obtained very similar results, with a maximum ELA difference of 25 m in year 2002, well below the estimated uncertainty of 80 m (1985–2018; Supplementary Table 3). The ablation zone area is calculated as the area below the ELA, whereas the firn area coincides with the accumulation zone area above the ELA.

**Observational data**. We use 1611 local (in situ) annual balance measurements covering the period 1967–2015 and collected at 101 sites (Fig. 1a) on Austre Brøggerbreen, Midtre Lovénbreen, Kongsvegen, and Holtedahlfonna glaciers in NW Svalbard[42,43]; Hansbreen glacier in SS sector[44]; AF ice cap[22] and Nordenskiöldbreen glacier in NE Svalbard[45]. Stake annual balance is estimated as the elevation difference between two consecutive end-of-summer surface heights (September). For a meaningful comparison, modelled SMB was integrated between September 15 of two consecutive years. The in situ data set is made available by the World Glacier Monitoring Service (WGMS) and was compiled by the University of Oslo, the Norwegian Polar Institute, the Polish Academy of Sciences, the University

of Uppsala and Utrecht University[27]. For consistency, we rejected four sites with >100 m height difference relative to the S0 Terreng DEM of Svalbard at 500 m spatial resolution. For comparison with stake measurements, we selected the downscaled grid cell with the smallest elevation bias among the closest pixel and its eight adjacent neighbours.

**Remotely sensed mass change**. We use a combination of GRACE mass change time series for the period 2002–2016[10] with elevation changes derived from ICESat (2003–2009) and CryoSat-2 (2010–2018). Following the method described in Gardner et al. (2013)[7] and Wouters et al. (2015)[46], ICESat records were grouped every 700 m along repeated ground tracks, whereas for CryoSat-2, neighbouring observations are collected within 1 km of each individual echo location. A model is fitted to these clusters of elevation observations in order to estimate the local surface topography and elevation rate at the central point, where outliers are removed in an iterative procedure. For full details, we refer the reader to Wouters et al. (2015)[46]. After estimating the local topography and elevation rate for the ICESat and CryoSat-2 periods, local elevation anomalies at the echo locations can be estimated by adding the elevation rate of the fitted model to the residuals. These anomalies are used to compute monthly volume anomalies for (individual) Svalbard ice caps. Elevation anomalies are parameterised as a function of absolute elevation using a third-order polynomial. The resulting fit is used to derive regional volume anomalies within 100 m elevation intervals, by multiplying the polynomial value at each interval's midpoint with the total glacier area within this elevation bin[1]. Finally, volume anomalies are converted to mass anomalies by assuming a constant density profile, using the density of ice below the ELA, and a density of 600 ± 250 kg m$^{-3}$ above the ELA[46].

**Bare ice area**. Annual modelled bare ice area is estimated for six sectors and the whole of Svalbard (Supplementary Tables 2 and 3) as the area of pixels showing a surface albedo ≤0.55 on the 11 km grid, bilinearly interpolated onto the 500 m ice mask, at least 2 days in that year. For comparison, we estimate annual bare ice extent using the broadband shortwave clear sky albedo data from the MCD43A3 MODIS 500-m 16-day albedo product. To eliminate spurious albedo records, erratic albedo grid cells were masked from the MODIS product (2000–2018) using the full bidirectional reflectance distribution function (BRDF) inversions. Valid MODIS records were classified as bare ice or snow-covered grid cells using an upper threshold for shortwave albedo of 0.55 (i.e. maximum albedo of bright bare ice under perennial firn). Subsequently, bare ice/snow cells were converted to annual bare ice extent if (i) the current pixel was classified as ice at least 5 days in that year (5th percentile), (ii) the pixel is located within the modelled ablation zone of that year (SMB < 0; 2000–2018), and (iii) the pixel is located below 700 m a.s.l., which is well above the long-term ELA of Svalbard (440 ± 80 m a.s.l. for 1985–2018) and individual sectors (up to 550 ± 65 m a.s.l. in NW; Supplementary Table 3). Even in extremely warm years such as 2003 and 2013, the Svalbard-wide

ELA (600 ± 80 m a.s.l.; Fig. 3b) remains below the selected elevation threshold. These criteria allow the elimination of pixels that represent meltwater lakes, superimposed ice and mountain range peaks at higher elevations as often found in the interior of Svalbard. The remaining masked pixels are filled on the basis of ice/snow recurrence for that cell: masked pixels are classified as bare ice if they expose bare ice more than 50% of the time in the period 2000–2018.

## Data availability

Data required to reproduce the tables and figures presented in the manuscript are freely available on PANGAEA https://doi.org/10.1594/PANGAEA.920984. These data include annual SMB and components downscaled to 500 m resolution (1958–2018): total precipitation (snowfall and rainfall), snowfall, runoff, melt, refreezing and retention, as well as summer (June–July–August) 2 m air temperature. Modelled (RACMO2.3; 1958–2018) and observed (MODIS; 2000–2018) bare ice area, and modelled ablation zone area (1958–2018) are also included. Daily downscaled SMB and components are available from the authors upon request and without conditions.

## Code availability

RACMO2.3 is presented in Noël et al. (2015)[29] and the statistical downscaling technique is described in Noël et al. (2016)[28].

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

## Acknowledgements

B.N. was funded by NWO VENI grant VI.Veni.192.019. C.L.J., C.H.R., W.J.B., and M.R.B. acknowledge support from the Polar Programme of the Netherlands Organiza-tion for Scientific Research (NWO/ALW) and the Netherlands Earth System Science Centre (NESSC). B.W. was funded by NWO VIDI grant 016.Vidi.171.063.

## Author contributions

B.N. prepared the manuscript, carried out the RACMO2.3 simulation and produced the downscaled dataset at 500 m. C.L.J. helped conducting and analysing the RACMO2.3 simulations. B.N., W.J.B and M.R.B. conceived the downscaling procedure and analysed the data. W.J.J.P., J.K., J.O.H., B.L. and C.H.R. provided the Svalbard in situ SMB dataset and the S0 Terreng DEM. S.L. processed the 500 m MODIS albedo product. B.W. produced and analysed the GRACE, ICESat and CryoSat-2 datasets. All authors commented on the manuscript.

## Competing interests

The authors declare no competing interests.
