## [Peer Review File · Nature Communications]

REVIEWER COMMENTS

Reviewer #1 (Remarks to the Author):

This paper investigates the mechanism behind the substantial mass loss for glaciers on Svalbard over the last few decades.

The importance of meltwater refreezing and the hypsometry of Svalbard glaciers is evaluated and the results are generally presented well and give useful insight into the very negative glacier mass balance in this region. Modelled results are compared with in situ and satellite measurements. This topic is one of great importance but where there is widespread confusion, even amongst glaciologists working in mountain or ice sheet environments, and thus is important to scientists in this and related fields.

Considering this journal is general, a considerable amount of prior knowledge is taken for granted in this paper. Terms and methods are mentioned (e.g. statistical downscaling; in situ measurements) with no references or additional text. This is acceptable in a specialist journal, but Nature Communications is read by a more general audience.

Apart from this most of the comments and suggested changes are cosmetic and are intended make the paper easier to understand.

The concept of "tipping point" is very applicable here, and considering previous work by the lead author, I was surprised it was not used. Reference to IPCC SROCC chapter 6 (or other useful reference) could be used.

Only one surge is mentioned. The cumulative effects of other surges should at least be considered.

Comments by line number:

1-3: First sentence – hard to understand, obtuse, and the second part of sentence doesn't follow on from first.

3: "efficient" necessary in abstract?

6: "hypsometry peak"- as the journal is non-specialist, this should be defined or non-specialist terminology used.

1-12: much of abstract is written as though one already has knowledge of the paper.

17: Define "arctic amplification" – especially as the reference referred to is not a published scientific article and the URL provided is obsolete and an error message in Norwegian pops up.

18-19: "Mountains locally peak at 19 1,717 m a.s.l" – "The highest point on Svalbard is 1717 m a.s.l" [this should also be checked as two different heights appear to be reported for this peak]

23-24: Include newer references, e.g. Wouters et al., 2019; Zemp et al., 2019

35: Both PDD and EBM are used only twice, hence acronyms are unnecessary and the phrase can be written out both times.

36: What does "a decin SMB" mean? Better to write – more negative surface mass balance (or lower winter accumulation, higher summer ablation as appropriate).

37: A reference (or brief explanation) of statistical downscaling should be given here, for readers unfamiliar with the technique.

48: "these northern ice caps" – ambiguous. I assume this refers just to ice caps on Svalbard, but could be interpreted to mean all ice caps in the north (Arctic).

(and 199-206): more information and/or references would be useful (both for non-glaciologists and for specialists) to explain what understood by in situ measurements. [or include in supplementary material if not room in main text]

65-67: were all other surges insignificant? How sure is calving term for 1958 – 1984, as large surges in this period may be undetected? Need more information.

73-82: It would be helpful to compare this with mass balance measurements on one or two glaciers with a record of several decades (although pattern not straightforward)

92: unnecessary detail – delete "both" and "at 700 hPa and"

93: insert "average" in front of equilibrium

93: the term "migrated" suggests a steady change. Suggest rewording this.

96: change "evidenced" to "as shown"
100: delete "now"
109: change to: (as shown by marked hypsometry peak.....)
109-110: delete "located at" (misleading when the elevation given is an average and interval)
144: change "oscillates" to "fluctuates"
147-148: change "two times more than the" to "twice as large as"
149: change "beyond" to "above"
152: replace "quasi-irreversible" by "irreversible" or "probably irreversible" as appropriate
161: change "input of" to "input to"
170: Are there also 40 snow/firn layers? If so, "The" should be deleted as misleading.
187: Replace "detailed" with "described".
188: Change "1.6 Gt yr-1 on average" to "an average of 1.6 Gt yr-1".
191: Insert "respectively" after Svalbard.
199: Delete "Here".
203: Delete "a"
215-216: "constant elevation rate" is an unusual expression and ambiguous. Please reword.
222: Insert reference for value of density used.
234: Is the value of 700m correct? It seems inconsistent with the rest of the text.
235: change "allow to discard" to "allow the elimination of"
236: change "as e.g." to "such as".

Figures

Figure 1: The inset map in 1b is too small to be useful. Delete the map and spell out the acronyms in the box instead. Write out SMB in the figure caption (SMB is defined in the text AFTER the first reference to figure 1). Change "The x-axis shows the glaciated area at each elevation level" to "The x-axis shows the glaciated area in each 100-m elevation band".

Supplementary Information

Line 27: Kohler is misspelled.

Fig 2c: The ICESat points are hard to see. The figure would be clearer if a darker colour was used for ICESat. Swapping the colours for ICESat/Cryosat-2 and SMB would help.

Fig 3a: There is a lot of useful information in this figure, but it's difficult to see. Perhaps simply making the graph taller would help.

Fig 3c: The key to symbols should be lines not dots.

Fig 4e: the black line delineating the ELA for 1985-2018 is hardly visible, except on Austfonna. A white line would be clearer.

- Miriam Jackson

Review of Noel et al., Svalbard glaciers

The paper is well-written, a valuable contribution to the growing Svalbard literature and it addresses a timely topic. The paper appears sound, although see general comment below which hampers evaluating this fully. My main concern (which can probably be easily addressed?) is that overall it is not quite clear what is new: obviously there have been many reconstructions and mass balance studies in the recent past which the authors cite, and the higher sensitivity due to the low elevation has been pointed out before. Perhaps it can be emphasized more what this study adds to this bulk of recent literature and the new results put in better context of the existing reconstructions which are barely mentioned.

Note, I make a number of comments where an issue occurs the first time but the comment may also hold for other places where I don't repeat the comment.

General

- 1) There are a number of terminology issues not consistent with Cogley et al., 2011 (Mass balance glossary), some of them make it difficult to evaluate if some of the methods are sound: SMB strictly speaking only refers to the surface balance (which ablation stakes measure) and not refreezing. The latter is included in the 'climatic mass balance'. It appears that what is modeled is the climatic balance and not the surface mass balance. This distinction is in particular important when it comes to comparison with in-situ stake measurements since these can only make surface mass balance and not the climatic balance. So, it is not clear if the evaluation was done comparing apples with pears (Supp Fig. 2), and thus if the evaluation is sound.
I recommend that the authors stick to Cogley et al., 2011.
- 2) It is also unclear how precipitation was treated. Rain fall does not contribute to mass balance (unless it refreezes). Do the authors mean snowfall when they say precipitation? Does the balance exclude rain that does not refreeze?
- 3) The term refreezing capacity is unclear. How is it defined? It appears that it is used in the sense of just 'refreezing' and not some sort of 'capacity' or ratio to pore space or so?
- 4) There is too many acronyms which are not necessary. There is generally enough space in the figures to spell them out and make the paper more readable to a broader audience.
- 5) Often in the text, the model results come across as 'facts'. It needs to be clearer that most results are modeled results.
- 6) It is not clear why the decline in the firn area and its associated capacity to store refrozen water is irreversible. If the climatic balance turned positive for sufficiently long time, the firn area would expand and thicken and with it allow more refreezing. So, the point of irreversibility appears speculative.

Detailed comments

- 7) Term 'Glaciated' is not used according to Cogley et al, 2011. → Glacierized or just glacier area

- 8) Line 1: this sentence is not appropriate for Nature readership. Isn't most important to convey that there is a lot of area below 450 m (or 500 m to round it). In theory a peak in the hypsometry can be extremely low but 90% of the area above. What matters is not just the peak but the distribution, so this may better be framed differently here and below.
- 9) Line 3: replace 'exist' by 'survive' since they do exist now
- 10) Overall the abstract is not very clear and seems not adequate for a broader readership; it's also not clear if this is observations or modeling results.
- 11) Line 8: 'dry climate'
- 12) All elevations should be m a.s.l., not just m
- 13) line 16, ref 3 is cited although an update by Farinotti et al 2019 exists
- 14) Line 21-22: again, the peak is not necessarily important, but how much area is below a certain elevation. E.g. at what elevation is X % of the area in these different regions.
- 15) Line 28
Terminology: SMB strictly speaking only refers to the surface balance (which ablation stakes measure) and not refreezing. The latter is included in the 'climatic mass balance'. Here it is the climatic balance
- 16) Line 49: vague: what kind of 'future warmer climate'.
- 17) Line 52/53: remove "yellow stars" and "in the ablation (blue)". This holds for other places in the manuscript as well. Please remove any reference to colors or symbols in the main text – this should only be in the captions. Just refer to the appropriate figure.
- 18) Line 53: what time periods do these 1611 measurements cover: are they annual balances, winter, summer, other periods?
- 19) Line 59 after "Fig 1a" to line 61: remove. The caption /figure shows these sectors
- 20) Line 75: I assume the authors mean 'net mass loss' since mass loss happens every year even in years with positive mass balance.
- 21) Line 79: 'confirms the large fluctuations': it is unclear what it confirms? Has this result been found by others and published and here your model comes up with the same result?
- 22) Line 88, not sure if 'mitigating' is the right word here
- 23) Line 98: why +62% this is unclear

- 24) Line 102: decline should be change if negative numbers follow
- 25) Line 161: remove 'In brief'
- 26) Methods: how is firn extent models?
- 27) 40 snow layers: how deep are subsurface processes modeled and how deep is the firn in Svalbard?
- 28) Modelled ELA: the method is unorthodox and can lead to very different results than when done as typically done, i.e. estimated from the mass balance gradient (where it's zero). Esp when ELAs are not rather straight lines, deviations can be large.
- 29) Line 157: bold statement without any evidence? Where does this come from
- 30) Overall a number of **figure captions** should be improved (I give some examples below). Often the first line reads like a campaign slogan. Best just to state what is shown rather than a 'conclusion like statement', e.g. 'Strong sensitivity' Also when there is a legend the colors of lines etc don't need to be repeated in the caption. Many caption can be shortened and thus become better readable.
- 31) **Figure 1:**
- a) color scale: would be better to reverse with positive numbers at top
 - b) yellow stars are almost impossible to see and the shape not at all. Perhaps just dots and a more distinguishable color
 - c) remove acronyms for plot b. They are not necessary and only make this figure hard to grasp – there is lots of space to spell out each region. GrIS: Greenland ice sheet, GIC: Greenland periphery or Greenland glaciers; NCAA: S Canadian Archipelago ... (the domain is clear from the map, 'Arctic' not needed). Same for Svalbard map, the full region names could be spelled out.
 - d) the inset map may be easier to read if it had a box
- 32) Figure 1 caption:
- a) add 'modeled' to caption in first line
 - b) 'elevation level': is this elevation band? Binning interval needs to be stated.
- 33) **Figure 2:**
- a) colors don't need to be repeated in the caption, there is a legend
 - b) legend is a bit confusing since all is mass balance, i.e. some form of mass change; and it's not immediately clear that there are 3 components shown based on this study and 3 sets of observational data sets. Perhaps the legend can be split into two columns?
 - c) green for both ICESat and CryoSat is not well visible
 - d) A color scheme suitable for color-blind people should be chosen
 - e) the Discharge curve is misleading since it looks like it is a mass gain (negative sea level rise). The problem is the lack of consistency in sign convention (gain +, losses -). This convention is followed for all curves but D. This needs to be fixed somehow best with using consistent sign convention, i.e. the D curve becomes a negative mass change.
 - f) Caption better: Modeled and observed mass changes and contr. ...
Not clear why 'Onset', it shows more than that.

34) Figure 3:

- a) spell out acronyms. There is enough space.
- b) caption: remove 'irreversible'
- c) caption: remove 'cumulative', it's not cumulative but the rate
- d) plot c: replace 'Integrated area' by Area
- e) plot c: the legend is confusing. Better (e.g. in 3 rows): Modeled ablation area, Modeled bare ice area, Observed bare ice area (MODIS)

Figure 4

- a) Remove JJA from axis label and put in caption: .. (a) 2 m summer (JJA) air temperature anomaly and ELA
- (b) remove 'change'
- also all other variables, just give the variable and no direction of change

Supplementary material

Figure S1: scale or coordinates missing

Figure S1 Legend:

- a) add 'modeled': Time series of modeled ...
- b) It's unclear why the right y-axis is only for the post-1985 mass loss. Doesn't the translation hold for the entire period?
- g) Can be added in the caption

Figure S2:

- a) Plot a: y-axis label: remove Downscaled and at 0.5 km, that can be in the caption
- b) Caption: Evaluation of what?
- c) Plot c: legend is confusing. All is mass balance, the difference is that the blue is modeled: replace 'Mass balance' by 'Modeled'
- d) Plot b: there should be some sort of multiplication symbol between the bias/rmse numbers and the 10^3
- e) Plot b: spell out legend acronyms; there is sufficient space

Figure S3:

- a) The acronyms are not needed and should be spelled out for easier readability

Figure S4:

- a) The caption: upward migration of the ELA seems misleading. I see the SMB and runoff change but no migration?
- b) The ELA for the 2 periods is not readable
- c) For clarity: 1985-2018 would be better in legend

Supple Table 1

- a) Comparison of what? Comparison of modeled and recent estimates.

- b) Explain PDD and EBM in caption, also WRF and MAR

Supple Table 2

- a) What's cap? How is it defined? The decimals seems not warranted, same for Ablation area
- b) What are the uncertainties for ELA?
- c) It might be better to call the ice masses just glaciers instead of ice caps, since I assume the table includes not only those ice masses that are from a morphological point an ice cap?

It would be useful if Figure 3a (main text) could also be expressed in specific units for comparison with mass change rates in other regions/studies. If the area was constant throughout the time period this could just be added as a second y-axis. If not perhaps another figure in the supplementary?

Response to reviewers

Dear reviewers, we would like to thank you for your constructive comments which greatly improved our manuscript. To facilitate readability of this document, our responses to reviewers are displayed in blue and modifications in the manuscript are highlighted in red. These suggested changes, together with additional minor corrections, are also displayed in red in the attached revised manuscript.

Reviewer #1: Miriam Jackson

This paper investigates the mechanism behind the substantial mass loss for glaciers on Svalbard over the last few decades. The importance of meltwater refreezing and the hypsometry of Svalbard glaciers is evaluated and the results are generally presented well and give useful insight into the very negative glacier mass balance in this region. Modelled results are compared with in situ and satellite measurements. This topic is one of great importance but where there is widespread confusion, even amongst glaciologists working in mountain or ice sheet environments, and thus is important to scientists in this and related fields. Thank you.

General comments:

1. Considering this journal is general, a considerable amount of prior knowledge is taken for granted in this paper. Terms and methods are mentioned (e.g. statistical downscaling; in situ measurements) with no references or additional text. This is acceptable in a specialist journal, but Nature Communications is read by a more general audience. We agree and revised the manuscript accordingly. See e.g. our responses to comments in L1-12, L37 and L51.
2. Apart from this most of the comments and suggested changes are cosmetic and are intended make the paper easier to understand. Thank you for thoroughly reading our manuscript, please find our corrections hereunder.
3. The concept of "tipping point" is very applicable here, and considering previous work by the lead author, I was surprised it was not used. Following the general comment #6 of reviewer #2, we removed the "tipping" or "irreversibility" aspect as this implies that recent mass loss will persist irrespective of future conditions even on the longer-term (centuries to millennia). Please, see our response to general comment #6 of reviewer #2.
4. Reference to IPCC SROCC chapter 6 (or other useful reference) could be used. The SROCC report is now referred to in the manuscript.
5. Only one surge is mentioned. The cumulative effects of other surges should at least be considered. Surging glaciers are common and widespread in Svalbard. Farnsworth et al. (2016) identified 708 Svalbard glaciers that have likely surged in the past. While the surging feature of Svalbard glaciers is well known (e.g. Hagen et al., 2003; Blaszczyk et al., 2009), surge events are poorly documented and understood. Only recent surges are described for a few glaciers in NW, SS and NE Svalbard (e.g. Sund et al., 2014; Sevestre et al., 2018; Nuth et al., 2019), and actual calving rates ($4.2 \pm 1.6 \text{ Gt yr}^{-1}$) were only estimated for the recent 2012-2013 surge of a major glacier in Austfonna (McMillan et al., 2014; Dunse et al., 2015). Finally, Blaszczyk et al. (2009) is the only study that provides Svalbard-wide calving flux estimates for 2000-2006 ($D = 6.8 \pm 1.7 \text{ Gt yr}^{-1}$), which we used in this paper. In addition, we included the post-2012 mass loss acceleration in Austfonna (Dunse et al., 2015). Unfortunately, based on the available literature, the cumulative effect of past surges cannot be accounted for. We inserted the following information in L32-37 as: "Surge-type glaciers strongly impact D and are widespread in Svalbard¹⁴, with more than 700 glaciers that likely surged in the past¹⁵. Although surge events can strongly influence mass loss locally¹⁶, these events are poorly understood and are only documented for a few glaciers¹⁷⁻¹⁹. Here we use a Svalbard-wide solid ice discharge estimate for the period 2000-2006¹³, complemented by an increase in D after the surge of a major Austfonna glacier in 2012-2013²⁰". See also our response to comment in L65-67.

Comments by line number:

1-3: First sentence – hard to understand, obtuse, and the second part of sentence doesn't follow on from first. We reformulated the abstract as: "Compared to other Arctic ice masses, Svalbard glaciers are low-elevated with flat interior accumulation areas, resulting in a marked peak in their current hypsometry (area-elevation distribution) at ~450 m above sea level. Since summer melt consistently

exceeds winter snowfall, these low-lying glaciers can only survive by refreezing a considerable fraction of surface melt and rain in the porous firn layer covering their accumulation zones. We use a high-resolution climate model to show that modest atmospheric warming in the mid-1980s forced the firn zone to retreat upward by ~100 m to coincide with the hypsometry peak. This led to a rapid areal reduction of firn cover available for refreezing, and strongly increased runoff from dark, bare ice areas, amplifying mass loss from all elevations. As the firn line fluctuates around the hypsometry peak in the current climate, Svalbard glaciers will continue to lose mass and show high sensitivity to temperature perturbations.”

3: “efficient” necessary in abstract? This is now removed. See also the previous comment in L1-3.

6: “hypsometry peak”- as the journal is non-specialist, this should be defined or nonspecialist terminology used. The term hypsometry is now defined as: “(area-elevation distribution)”.

1-12: much of abstract is written as though one already has knowledge of the paper.
We reformulated the abstract, see our response to comment in L1-3.

17: Define “arctic amplification” – especially as the reference referred to is not a published scientific article and the URL provided is obsolete and an error message in Norwegian pops up. We define Arctic Amplification in L15-17 following Chapter 3 of the SROCC report: “As a result of Arctic Amplification³, in which Arctic warming over the last two decades was twice the global average⁴, and being situated at the edge of retreating Arctic sea ice, Svalbard ice caps experience among the fastest warming on Earth.”. The corrupted URL has been fixed.

18-19: “Mountains locally peak at 19 1,717 m a.s.l.” – “The highest point on Svalbard is 1717 m a.s.l.” [this should also be checked as two different heights appear to be reported for this peak]
The value of 1,717 m a.s.l. is from Van Pelt et al. (2019). The exact value is not essential and we decided to round it to ~1700 m a.s.l. in L18-21 as: “The highest elevation on Svalbard is approximately 1,700 m a.s.l. (above sea level), but the glacier hypsometry (area-elevation distribution) peaks at ~450 m a.s.l., compared to 800-1,400 m a.s.l. for ice caps in Greenland, Arctic Canada and Iceland (Fig. 1b).”.

23-24: Include newer references, e.g. Wouters et al., 2019; Zemp et al., 2019. Done. We also included the recent work of Schuler et al. (2020) in L46-47: “Similar conclusions were drawn by upscaling in situ SMB measurements to all Svalbard land ice¹², but little remains known about the temporal and spatial variability of the surface mass loss.” ; and added their mass balance estimates in Table S1 in the Supplementary Material.

35: Both PDD and EBM are used only twice, hence acronyms are unnecessary and the phrase can be written out both times. Indeed, thank you. Acronyms have been removed accordingly.

36: What does “a decline SMB” mean? Better to write – more negative surface mass balance (or lower winter accumulation, higher summer ablation as appropriate). We reformulated as: “increase in summer ablation (Table S1)”.

37: A reference (or brief explanation) of statistical downscaling should be given here, for readers unfamiliar with the technique. We now briefly explain the downscaling technique in L53-56 as: “The method primarily corrects daily melt and runoff for elevation biases on the relatively coarse RACMO2.3 model grid using elevation gradients, and for underestimated ice albedo using remote sensing measurements²⁸ (see Methods).”.

48: “these northern ice caps” – ambiguous. I assume this refers just to ice caps on Svalbard, but could be interpreted to mean all ice caps in the north (Arctic). We reformulated as: “Svalbard ice caps”.

51 (and 199-206): more information and/or references would be useful (both for non-glaciologists and for specialists) to explain what understood by in situ measurements. [or include in supplementary material if not room in main text]. We reformulated L67-69 as: “The SMB product is evaluated using

1,611 local (in situ) annual balance measurements from 101 sites (Fig. 1a) collected in the ablation and accumulation zones of Svalbard glaciers over the period 1967-2015 (see Methods; Fig. S2a).” and described the in situ sites and measurements in the Method section in L232-238 as: “We use 1,611 local (in situ) annual balance measurements covering the period 1967-2015 and collected at 101 sites (Fig. 1a) on Austre Brøggerbreen, Midtre Lovénbreen, Kongsvegen and Holtedahlfonna glaciers in NW Svalbard^{42,43}; Hansbreen glacier in SS sector⁴⁴; Austfonna ice cap²² and Nordenskiöldbreen glacier in NE Svalbard⁴⁵. Stake annual balance is estimated as the elevation difference between two consecutive end-of-summer surface heights (September). For a meaningful comparison, modelled SMB was integrated between September 15 of two consecutive years.”. Appropriate references to the data sets have also been inserted accordingly.

65-67: were all other surges insignificant? How sure is calving term for 1958 – 1984, as large surges in this period may be undetected? Need more information. Surges are common in Svalbard but poorly documented with no calving estimates. Blaszczyk et al. (2009) provide the only Svalbard-wide estimate of solid ice discharge for the period 2000-2006 ($6.8 \pm 1.7 \text{ Gt yr}^{-1}$). To the authors’ knowledge, Dunse et al. (2015) is the only study that quantifies mass loss from the major surge event in Austfonna in 2012-2013 ($4.2 \pm 1.6 \text{ Gt yr}^{-1}$). Therefore, we assumed that the 2000-2006 solid ice discharge flux of Blaszczyk et al. (2009) was valid for the whole study period, and included the recent contribution of Austfonna after 2012 (Dunse et al., 2015) in very good agreement with remote sensing. We clarified this in L83-89 as: “We assume that solid ice discharge estimate for 2000-2006 ($D = 6.8 \pm 1.8 \text{ Gt yr}^{-1}$)¹³ is valid for the whole study period (1958-2018). In line with Dunse et al. (2015)²⁰, we increase solid ice discharge by $4.2 \pm 1.6 \text{ Gt yr}^{-1}$ from 2012 onwards, following the surge of a major Austfonna outlet glacier. Combining this with the downscaled SMB product, we reconstruct the mass change of Svalbard glaciers over the last six decades (Fig. 2). The modelled mass change is obtained by integrating both SMB and D in time starting from zero in 1958. Our reconstruction agrees very well with remote sensing records from GRACE [...]”.

73-82: It would be helpful to compare this with mass balance measurements on one or two glaciers with a record of several decades (although pattern not straightforward). Compared to previous studies we are aware of, the model evaluation presented here is already very comprehensive and agrees well with in situ/remotely sensed observations. Therefore, we feel that that additional comparisons are not necessary to further support the quality of our product.

92: unnecessary detail – delete “both” and “at 700 hPa and”. Done. 93: insert “average” in front of equilibrium. Done. 93: the term “migrated” suggests a steady change. Suggest rewording this. We replaced “migrated” by “moved”. 96: change “evidenced” to “as shown”. Done. 100: delete “now”. Done. 109: change to: (as shown by marked hypsometry peak.....). Done. 109-110: delete “located at” (misleading when the elevation given is an average and interval). Done. 144: change “oscillates” to “fluctuates”. Done. 147-148: change “two times more than the” to “twice as large as”. Done. 149: change “beyond” to “above”. Done. 152: replace “quasi-irreversible” by “irreversible” or “probably irreversible” as appropriate. We reformulated in L173-174 as: “We conclude that the post-1985 decline in refreezing capacity will persist under continued warming: [...]”. See also our response to general comment #3. 161: change “input of” to “input to”. Done. 170: Are there also 40 snow/firn layers? If so, “The” should be deleted as misleading. Yes, RACMO2 has 40 snow layers. We reformulated as: “The model also includes 40 active snow layers [...]”. 187: Replace “detailed” with “described”. Done. 188: Change “1.6 Gt yr-1 on average” to “an average of 1.6 Gt yr-1”. Done. 191: Insert “respectively” after Svalbard. Done. 199: Delete “Here”. Done. 203: Delete “a”. Done. 215-216: “constant elevation rate” is an unusual expression and ambiguous. Please reword. We reformulated as: “[...] estimated by adding the elevation rate of the fitted model to the residuals.” 222: Insert reference for value of density used. We refer to Wouters et al. (2015) in which the technique is described in detail. 235: change “allow to discard” to “allow the elimination of”. Done. 236: change “as e.g.” to “such as”. Done.

234: Is the value of 700m correct? It seems inconsistent with the rest of the text. As shown in Fig. 3b, long-term ELA of Svalbard (and individual sectors; see Table S3) remains well below 700 m a.s.l. Only in extremely warm summers (2003 and 2013; Fig. 3b) does the Svalbard-wide ELA reach elevations of $600 \pm 80 \text{ m a.s.l.}$ In addition, Fig. 3c highlights that the ELA remains above the bare ice zone as the

upper ablation zone also includes a narrow band exposing superimposed ice. Therefore, an upper elevation threshold of 700 m was judged appropriate to eliminate spurious snow free pixels in our MODIS albedo product that result from e.g. superimposed ice, meltwater lakes or bare rocks. This is now clarified in L271-274 as: “and iii) the pixel is located below 700 m a.s.l., which is well above the long-term ELA of Svalbard (440 ± 80 m a.s.l. for 1985-2018) and individual sectors (up to 550 ± 65 m a.s.l. in NW; Table S3). Even in extremely warm years such as 2003 and 2013, the Svalbard-wide ELA (600 ± 80 m a.s.l.; Fig. 3b) remains below the selected elevation threshold.”.

Comments on Figures:

Figure 1: The inset map in 1b is too small to be useful. Delete the map and spell out the acronyms in the box instead. **Done.** Write out SMB in the figure caption (SMB is defined in the text AFTER the first reference to figure 1). **Done.** Change “The x-axis shows the glaciated area at each elevation level” to “The x-axis shows the glaciated area in each 100-m elevation band”. **Done.**

Supplementary Information

Line 27: Kohler is misspelled. **Thank you for pointing that out! This is corrected.** Fig 2c: The ICESat points are hard to see. The figure would be clearer if a darker colour was used for ICESat. Swapping the colours for ICESat/Cryosat-2 and SMB would help. **We prefer not swapping colours between ICESat/CryoSat-2 and SMB to be consistent with Fig. S2c.** For clarity, ICESat data are now shown in light blue and we increased the marker size. For consistency, similar changes were applied to Fig. S2c. Fig 3a: There is a lot of useful information in this figure, but it’s difficult to see. Perhaps simply making the graph taller would help. **Enlarging Fig. 3a is difficult given the other three time series below and the second y-axis requested by reviewer #2. We hope that it is now better readable.** Fig 3c: The key to symbols should be lines not dots. **Done.** Fig 4e: the black line delineating the ELA for 1985-2018 is hardly visible, except on Austfonna. A white line would be clearer. **As suggested, we used a white line to outline the ELA but were not satisfied with the end result (see attached figure hereunder), and decided to leave Fig. 4e as is. If judged necessary by the editor, we can revise Fig. 4e.**

Reviewer #2: Anonymous

The paper is well-written, a valuable contribution to the growing Svalbard literature and it addresses a timely topic. The paper appears sound, although see general comment below which hampers evaluating this fully. Thank you.

My main concern (which can probably be easily addressed?) is that overall it is not quite clear what is new: obviously there have been many reconstructions and mass balance studies in the recent past which the authors cite, and the higher sensitivity due to the low elevation has been pointed out before. We present a new SMB data set for Svalbard that, for the first time, reconstructs realistic mass balance in space and time at high spatial (500 m) and temporal (daily) resolution over the last six decades. We demonstrate that the results are in close agreement with recent remote sensing estimates and long-term mass change from previous studies (Table S1). Our results highlight that Svalbard glaciers have experienced large spatial and temporal mass loss variability since the mid-1980s and enable a process-based interpretation, namely rapid fluctuations of the firn line around the peak in glacier hypsometry. To the authors' knowledge, no previous studies have quantified these processes in similar detail. This is stressed in L58-65 as: "Combined with discharge estimates^{13,20}, our high-resolution SMB product enables us to estimate the spatially and temporally varying mass balance of Svalbard glaciers over the last six decades, including the high mass loss variability starting in the mid-1980s. We show that a modest atmospheric warming of 0.5°C in the mid-1980s was sufficient to raise the firn line to the hypsometry peak at ~450 m a.s.l., exposing large parts of the accumulation area to increased melt. The subsequent loss of refreezing capacity, i.e. the fraction of rain and meltwater retained or refrozen in firn (see Methods), implies that Svalbard ice caps can no longer be sustained when the current climate persists or further warming occurs."

Perhaps it can be emphasized more what this study adds to this bulk of recent literature and the new results put in better context of the existing reconstructions which are barely mentioned. In the introduction, we provide a comprehensive collection of recent studies that estimated the (surface) mass balance of Svalbard using various techniques (e.g. regional climate models, positive degree day and energy balance models, GRACE, geodetic and in situ measurements; see L37-46 in the revised manuscript). Addressing the comments in L23-24 and general comment #5 of Reviewer #1, we now also include the recent work of Zemp et al. (2019) and Schuler et al. (2020), and added a dedicated paragraph on the contribution of solid ice discharge from surge-type glaciers to the mass balance of Svalbard (see L27-36), including the work of Sund et al. (2014), Farnsworth et al. (2016), Sevestre et al. (2018) and Nuth et al. (2019). In addition, we provide a comprehensive assessment of mass balance estimates derived from 14 studies, and compare them with our reconstructed product in Table S1. We deem that our approach keeps the manuscript concise and focused on our main message. A detailed assessment of techniques previously used to estimate (surface) mass balance of Svalbard is available in e.g. the recent Review Article of Schuler et al. (2020).

Note, I make a number of comments where an issue occurs the first time but the comment may also hold for other places where I don't repeat the comment. These remarks have been accounted for across the revised manuscript.

General comments:

1) There are a number of terminology issues not consistent with Cogley et al., 2011 (Mass balance glossary), some of them make it difficult to evaluate if some of the methods are sound: SMB strictly speaking only refers to the surface balance (which ablation stakes measure) and not refreezing. The latter is included in the 'climatic mass balance'. It appears that what is modeled is the climatic balance and not the surface mass balance.

Indeed, our use of 'surface mass balance' (SMB) includes 'internal accumulation' (refreezing and retention), and conforms to a quantity that is formally referred to as 'climatic mass balance' in Cogley et al. (2011). However, the use of SMB conforms to previous studies for Canadian ice caps, Greenland ice caps and the Greenland ice sheet (Noël et al., 2017, 2018, 2019) and Svalbard (Van Pelt et al., 2019; Lang et al., 2015). Our way of using SMB is also commonly found in widely cited ice sheet mass balance studies where the mass balance definition \$MB = SMB - D\$ (solid ice discharge) is used, e.g. Shepherd et al. (2020; Nature), Mougnot et al. (2019; PNAS). Given that our slightly different way of using 'surface mass balance' is widely used and accepted, we feel its current use is acceptable as long

as it is clearly stated and defined in the manuscript, i.e. in the revised Methods section in L187-193 we now state: “In this study we refer to ‘surface mass balance’ (SMB) as both the local (kg m^{-2}) and spatially integrated (Gt yr^{-1}) sum of:

$$\text{SMB} = \text{PR} - \text{RU} - \text{SU} - \text{ER} \quad (1)$$

where PR represents total precipitation including snowfall (SF) and rainfall (RA), RU meltwater runoff, SU total sublimation and ER the erosion from drifting snow. Liquid water from rain and melt (ME) that is not retained/refrozen in firn (RF) contributes to runoff:

$$\text{RU} = \text{ME} + \text{RA} - \text{RF} \quad (2)$$

Note that in Cogley and others (2011)³⁷, the local quantity that includes ‘internal accumulation’ from refreezing/retention is referred to as ‘climatic mass balance’.

This distinction is in particular important when it comes to comparison with in-situ stake measurements since these can only make surface mass balance and not the climatic balance. So, it is not clear if the evaluation was done comparing apples with pears (Supp Fig. 2), and thus if the evaluation is sound. I recommend that the authors stick to Cogley et al., 2011. We agree that stake measurements from the percolation area (lower accumulation zone) and the interior accumulation zone do not include internal accumulation. To address this, we repeated the evaluation shown in Fig. S2a and compared in situ measurements to modelled “SMB minus RF”, where RF accounts for refreezing and retention, for sites located in the accumulation zone i.e. above the long-term Svalbard-wide ELA of 440 m a.s.l. (see Table S3). Not accounting for internal accumulation (i.e. refreezing) in the accumulation zone, we obtain very similar results to those shown in Fig. S2a, with a slightly larger RMSE (+50 mm w.e.) and smaller R^2 (-0.07). It is also important to note that in situ measurements suffer from relatively large uncertainties. Therefore, we deem that the evaluation shown in Fig. S2 remains valid and clarified this in L71-75: “Unlike the downscaled SMB product, stake measurements in the accumulation zone do not include internal accumulation from the refreezing of melt and rain (see Methods). Ignoring internal accumulation when comparing the model to stake measurements located in the accumulation zone leads to a small RMSE increase of $\sim 50 \text{ mm w.e. yr}^{-1}$ ”.

2) It is also unclear how precipitation was treated. Rainfall does not contribute to mass balance (unless it refreezes). Do the authors mean snowfall when they say precipitation? This is indeed confusing, by precipitation we mean “total precipitation” (PR) including both snow and rainfall. The revised manuscript now refers to “total precipitation” where appropriate. Does the balance exclude rain that does not refreeze? No rain that does not refreeze is considered part of surface runoff as in Eq. (2) above.

3) The term refreezing capacity is unclear. How is it defined? It appears that it is used in the sense of just ‘refreezing’ and not some sort of ‘capacity’ or ratio to pore space or so? Firn refreezing capacity is estimated as the fraction of rain and meltwater that is retained or refrozen in snow. This is now clarified in L63-65 as: “The subsequent loss of refreezing capacity, i.e. the fraction of rain and meltwater retained or refrozen in firn (see Methods), implies that Svalbard ice caps can no longer be sustained when the current climate persists or further warming occurs.”. The Methods section also

includes the following in L193-195: “Firn refreezing capacity (RFcap), i.e. the fraction of rain and meltwater effectively retained or refrozen in firn, is estimated as, $RFcap = RF / (ME + RA)$ (3)”.

4) There is too many acronyms which are not necessary. There is generally enough space in the figures to spell them out and make the paper more readable to a broader audience. The manuscript has been revised accordingly.

5) Often in the text, the model results come across as ‘facts’. It needs to be clearer that most results are modeled results. We made an effort to clarify that results are based on model outputs in the revised manuscript.

6) It is not clear why the decline in the firn area and its associated capacity to store refrozen water is irreversible. If the climatic balance turned positive for sufficiently long time, the firn area would expand and thicken and with it allow more refreezing. So, the point of irreversibility appears speculative. Our use of the term ‘irreversible’ indeed assumed persistence of current conditions or further warming, and not a future transition to colder conditions. So we agree that the loss of refreezing capacity might not be “irreversible” on centennial to millennial time scales. We therefore decided to tone down the “irreversibility” claim and reformulated L63-65 as: “The subsequent loss of refreezing capacity, i.e. the fraction of rain and meltwater retained or refrozen in firn (see Methods), implies that Svalbard ice caps can no longer be sustained when the current climate persists or further warming occurs.”, in L173-175 as: “We conclude that the post-1985 decline in firn refreezing capacity will persist under continued warming: a temporary [...] recovery of the refreezing capacity (Fig. 3d).”.

Detailed comments:

7) Term ‘Glaciated’ is not used according to Cogley et al, 2011. → Glacierized or just glacier area We replaced “glaciated area” by “glacier area” where appropriate.

8) Line 1: this sentence is not appropriate for Nature readership. Isn’t most important to convey that there is a lot of area below 450 m (or 500 m to round it). In theory a peak in the hypsometry can be extremely low but 90% of the area above. What matters is not just the peak but the distribution, so this may better be framed differently here and below. We reformulated the abstract as: “Compared to other Arctic ice masses, Svalbard glaciers are low-elevated with flat interior accumulation areas, resulting in a marked peak in their current hypsometry (area-elevation distribution) at ~450 m above sea level. Since summer melt consistently exceeds winter snowfall, these low-lying glaciers can only survive by refreezing a considerable fraction of surface melt and rain in the porous firn layer covering their accumulation zones. We use a high-resolution climate model to show that modest atmospheric warming in the mid-1980s forced the firn zone to retreat upward by ~100 m to coincide with the hypsometry peak. This led to a rapid areal reduction of firn cover available for refreezing, and strongly increased runoff from dark, bare ice areas, amplifying mass loss from all elevations. As the firn line fluctuates around the hypsometry peak in the current climate, Svalbard glaciers will continue to lose mass and show high sensitivity to temperature perturbations.” Based on the data of Fig. 1b, we estimated that about 60% of the glacier area was below 450 m a.s.l. See also our response to comment #14.

9) Line 3: replace ‘exist’ by ‘survive’ since the do exist now. Done.

10) Overall the abstract is not very clear and seems not adequate for a broader readership; it’s also not clear if this is observations or modeling results. See our response to comment #8 above.

11) Line 8: ‘dry climate’. This is removed in the revised manuscript.

12) All elevations should be m a.s.l., not just m. Done.

13) line 16, ref 3 is cited although an update by Farinotti et al 2019 exists. Thank you for pointing that out! We updated the reference and associated values in L13-15: “[...] they contain \$7,740 \pm 1,940 \text{ km}^3\$ (or Gigaton; Gt) of ice, sufficient to raise global sea level by \$1.7 \pm 0.5 \text{ cm}\$ if totally melted².”.

14) Line 21-22: again, the peak is not necessarily important, but how much area is below a certain elevation. E.g. at what elevation is X % of the area in these different regions. To reflect this, we added L21-22 as: “About 60% of the total glacier area of Svalbard is located below that hypsometry peak.”.

15) Line 28 Terminology: SMB strictly speaking only refers to the surface balance (which ablation stakes measure) and not refreezing. The latter is included in the ‘climatic mass balance’. Here it is the climatic balance. See our response to general comment #1.

16) Line 49: vague: what kind of ‘future warmer climate’. We meant if climate warming continues at the current rate. We reformulated L63-65 as: “The subsequent loss of refreezing capacity, i.e. the fraction of rain and meltwater retained or refrozen in firn (see Methods), implies that Svalbard ice caps can no longer be sustained when the current climate persists or further warming occurs.”

17) Line 52/53: remove “yellow stars” and “in the ablation (blue)”. This holds for other places in the manuscript as well. Please remove any reference to colors or symbols in the main text – this should only be in the captions. Just refer to the appropriate figure. We modified the manuscript accordingly.

18) Line 53: what time periods do these 1611 measurements cover: are they annual balances, winter, summer, other periods? We used 1,611 annual balance measurements covering the period 1967-2015. We reformulated L67-69 as: “The SMB product is evaluated using 1,611 local (in situ) annual balance measurements from 101 sites (Fig. 1a) collected in the ablation and accumulation zones of Svalbard glaciers over the period 1967-2015 (see Methods; Fig. S2a).” and described the in situ sites and measurements in the Method section in L232-238 as: “We use 1,611 local (in situ) annual balance measurements covering the period 1967-2015 and collected at 101 sites (Fig. 1a) on Austre Brøggerbreen, Midtre Lovénbreen, Kongsvegen and Holtedahlfonna glaciers in NW Svalbard ^{42, 43}; Hansbreen glacier in SS sector ⁴⁴; Austfonna ice cap ²² and Nordenskiöldbreen glacier in NE Svalbard ⁴⁵. Stake annual balance is estimated as the elevation difference between two consecutive end-of-summer surface heights (September). For a meaningful comparison, modelled SMB was integrated between September 15 of two consecutive years.”. Appropriate references to the data sets have also been inserted accordingly.

19) Line 59 after “Fig 1a” to line 61: remove. The caption /figure shows these sectors. For clarity, we prefer listing each sector and their acronym once, as they will be heavily used in the Ablation zone and firn line reread and Discussion sections. We rephrased L79-81 as: “To that end, we divide Svalbard in six sectors (Fig. 1a) namely Northwest (NW), Northeast (NE), Vestfonna (VF), Austfonna (AF), Barentsøya and Edgeøya (BE), and South Spitsbergen (SS).”.

20) Line 75: I assume the authors mean ‘net mass loss’ since mass loss happens every year even in years with positive mass balance. Indeed, thank you this is corrected.

21) Line 79: ‘confirms the large fluctuations’: it is unclear what it confirms? Has this result been found by others and published and here your model comes up with the same result? ‘Confirms’ is related to observed remotely sensed mass changes. This is now clarified in L100-102 as: “Both remote sensing data and our reconstruction show that Svalbard glaciers have experienced mass loss since the mid-1980s, including the pause between 2005-2012.”.

22) Line 88, not sure if ‘mitigating’ is the right word here. We replaced by “reducing”.

23) Line 98: why +62% this is unclear. We agree that this is confusing and removed it altogether.

24) Line 102: decline should be change if negative numbers follow. We reformulated L124 as: “[...] similar decline in refreezing capacity, ranging from 22% in NW to 36% in BE sectors”.

25) Line 161: remove ‘In brief’. Done.

26) Methods: how is firn extent modeled? The firn line corresponds to the interface between the

accumulation and ablation zone (i.e. ELA). The firn area is defined as the accumulation zone area, i.e. the residual of the ablation zone area listed in Tables S2 and S3. This is clarified in L230-231: “The ablation zone area is calculated as the area below the ELA, whereas the firn area coincides with the accumulation zone area above the ELA.”.

27) 40 snow layers: how deep are subsurface processes modeled and how deep is the firn in Svalbard? The snow layer in RACMO2 can be as deep as 30 to 40 m in Svalbard. This is now clarified in L198 as: “In RACMO2.3 Svalbard firn can be 30 to 40 m deep locally.”.

28) Modelled ELA: the method is unorthodox and can lead to very different results than when done as typically done, i.e. estimated from the mass balance gradient (where it's zero). Esp when ELAs are not rather straight lines, deviations can be large. To ensure that our method realistically samples the evolution of the ELA (and its uncertainty) for individual glaciers and Svalbard-wide, we repeated the procedure using different thresholds i.e. ± 5 , ± 25 , ± 75 and ± 100 mm w.e. yr^{-1} . We obtained very similar results (see Figure below). The threshold of ± 50 mm w.e. yr^{-1} was selected as a trade-off between sufficient available pixels to estimate the ELA and a relatively low SMB threshold to avoid overestimation (threshold > 50 mm w.e.) / underestimation (threshold < 50 mm w.e.) of the ELA. A maximum height difference of 25 m is obtained for the 5 mm w.e. yr^{-1} threshold in 2002, which is significantly less than the estimated uncertainty of 80 m i.e. 1 standard deviation of the period 1985-2018. This is now clarified in L227-230 as: “We repeated the procedure using various thresholds ranging from 5 to 100 mm w.e. and obtained very similar results with a maximum ELA difference of 25 m in year 2002, well below the estimated uncertainty of 80 m (1985-2018; Table S3)”.

29) Line 157: bold statement without any evidence? Where does this come from? We used the “modelled” mass loss rate of the period 2013-2018 of 19.4 ± 3.4 Gt yr^{-1} , i.e. SMB (-8.4 Gt yr^{-1}) minus calving from combined Blaszczyk et al. (2009; 6.8 Gt yr^{-1}) and Dunse et al. (2015; 4.2 Gt yr^{-1}), and the previous ice volume estimates of 5200 - 7300 km³ (or Gt) from Fürst et al. (2018) to derive an early-late timing of Svalbard deglaciation. Early estimate of $5200/19.4 = 268$ or about 250 years; late estimate of $7300/19.4 = 376$ or about 400 years. Using the updated estimate of Farinotti et al. (2019), we obtain: $7470/19.4 = 385$ or approximately 400 years. This is now updated in L175-177 as: “At the current mass loss rate (19.4 ± 3.4 Gt yr^{-1} for 2013-2018), Svalbard glaciers would completely melt within the next 400 years.”

30) Overall a number of figure captions should be improved (I give some examples below). Often the first line reads like a campaign slogan. Best just to state what is shown rather than a ‘conclusion like

statement', e.g. 'Strong sensitivity ...' We modified the caption of Fig. 2 as: "Cumulative mass change of Svalbard glaciers and contribution to sea level rise."; Fig. 3 as: "Ablation zone expansion and reduced refreezing capacity."; Fig. 4 as: "Sensitivity of Svalbard refreezing capacity to atmospheric warming."; Fig. 5 as: "Ablation zone expansion in summer 2013."; Fig. S2 as: "Model evaluation using in situ and remote sensing measurements."; Fig. S4 as: "Ablation zone expansion and runoff change."; Table S1 as: "Model evaluation using recent mass change estimates."

Also when there is a legend the colors of lines etc don't need to be repeated in the caption. Many caption can be shortened and thus become better readable. We modified the captions accordingly.

Comments on Figures:

31) Figure 1: a) color scale: would be better to reverse with positive numbers at top. Done. b) yellow stars are almost impossible to see and the shape not at all. Perhaps just dots and a more distinguishable color. We now use orange dots. c) remove acronyms for plot b. They are not necessary and only make this figure hard to grasp – there is lots of space to spell out each region. GrIS: Greenland ice sheet, GIC: Greenland periphery or Greenland glaciers; NCAA: S Canadian Archipelago ... (the domain is clear from the map, 'Arctic' not needed). Same for Svalbard map, the full region names could be spelled out. Done. d) the inset map may be easier to read if it had a box. This was removed following reviewer #1's suggestions.

32) Figure 1 caption: a) add 'modeled' to caption in first line. Done. b) 'elevation level': is this elevation band? Binning interval needs to be stated. This was revised following reviewer #1's comment as: "The x-axis shows the glacier area in each 100 m elevation band as a fraction of the total ice area of that region (%)."

33) Figure 2: a) colors don't need to be repeated in the caption, there is a legend. Done. b) legend is a bit confusing since all is mass balance, i.e. some form of mass change; and it's not immediately clear that there are 3 components shown based on this study and 3 sets of observational data sets. Perhaps the legend can be split into two columns? Done. c) green for both ICESat and CryoSat is not well visible. For clarity, ICESat data are now shown in light blue. This also holds for Fig. S2c. d) A color scheme suitable for color-blind people should be chosen. We deem that the palette used is sufficiently clear and contrasted. In addition, these colours are consistent with our previous publications e.g. Noël et al. (2017). e) the Discharge curve is misleading since it looks like it is a mass gain (negative sea level rise). The problem is the lack of consistency in sign convention (gain +, losses -). This convention is followed for all curves but D. This needs to be fixed somehow best with using consistent sign convention, i.e. the D curve becomes a negative mass change. As mentioned in the caption, only the "reconstructed mass balance" (MB) is converted into sea level rise equivalent. As D cannot be negative, our use of signs is physically consistent: SMB and D are both positive, only mass balance becomes negative as $MB = SMB - D$: "The right y-axis translates Svalbard cumulative mass balance into global sea level rise equivalent.". In addition, showing D as negative would make the figure more confusing, with 5 different time series (MB, D, GRACE, ICESat, CryoSat-2) confined in the lower part of the graph. In line with our previous work (e.g. Van den Broeke et al., 2016; The Cryosphere), we decided to leave it as is. f) Caption better: Modeled and observed mass changes and contr. ... Not clear why 'Onset', it shows more than that. We reformulated as: "Cumulative mass change of Svalbard glaciers and contribution to sea level rise.". a) add 'modeled': Time series of modeled ... We reformulated as: "Time series of monthly cumulative modelled SMB, measured cumulative solid ice discharge (D)^{11,12} and reconstructed cumulative mass balance (MB = SMB minus D) for the period 1958-2018. Observed mass change derived from GRACE (2002-2016), ICESat (2003-2009) and CryoSat-2 (2010-2018) are also shown.". b) It's unclear why the right y-axis is only for the post-1985 mass loss. Doesn't the translation hold for the entire period? Indeed, we reformulated as: "[...] reconstructed cumulative mass balance (MB = SMB minus D) for the period 1958-2018."

34) Figure 3: a) spell out acronyms. There is enough space. Done. b) caption: remove 'irreversible'. Done. c) caption: remove 'cumulative', it's not cumulative but the rate. Done. d) plot c: replace 'Integrated area' by Area. Done. e) plot c: the legend is confusing. Better (e.g. in 3 rows): Modeled ablation area, Modeled bare ice area, Observed bare ice area (MODIS). Done. It would be useful if Figure 3a (main text) could also be expressed in specific units for comparison with mass change rates in other regions/studies. If the area was constant throughout the time period this could just be added

as a second y-axis. If not perhaps another figure in the supplementary? As suggested, we now include a second y-axis converting Gt yr⁻¹ into m w.e. yr⁻¹.

35) Figure 4 a) Remove JJA from axis label and put in caption: .. (a) 2 m summer (JJA) air temperature anomaly and ELA. Done. (b) remove 'change' also all other variables, just give the variable and no direction of change. This was corrected accordingly.

Supplementary material

Figure S1: scale or coordinates missing. Scale added.

Figure S2: a) Plot a: y-axis label: remove Downscaled and at 0.5 km, that can be in the caption. We prefer keeping the label as is for clarity. b) Caption: Evaluation of what? We reformulated as: "Model evaluation [...]". c) Plot c: legend is confusing. All is mass balance, the difference is that the blue is modeled: replace 'Mass balance' by 'Modeled'. The mass balance is not "stricto sensu" modelled since it is estimated as the difference between modelled SMB and measured Discharge. We reformulated as: "reconstructed mass balance". d) Plot b: there should some sort of multiplication symbol between the bias/rmse numbers and the 10³. Done. e) Plot b: spell out legend acronyms; there is sufficient space. Done.

Figure S3: a) The acronyms are not needed and should be spelled out for easier readability. Done.

Figure S4: a) The caption: upward migration of the ELA seems misleading. I see the SMB and runoff change but no migration? We rephrased as: "Ablation zone expansion and runoff change." b) The ELA for the 2 periods is not readable. Given the shape and size of Svalbard glaciers we could not improve the representation of the ELAs and left these as were. We deem that major ELA changes are well visible. c) For clarity: 1985-2018 would be better in legend. Done.

Supple Table 1 a) Comparison of what? Comparison of modeled and recent estimates. We reformulated as: "Model evaluation [...]". b) Explain PDD and EBM in caption, also WRF and MAR. Done.

Supple Table 2 a) What's cap? How is it defined? The decimals seems not warranted, same for Ablation area. We replaced "cap." by "capacity" and kept the decimals. b) What are the uncertainties for ELA? As mentioned in the Methods, ELA uncertainty is estimated as one standard deviation of the Svalbard-wide ELA (or individual sectors) for each period i.e. 1958-1984 and 1985-2018. This is now clarified in the caption as: "ELA uncertainty is estimated as one standard deviation of the period 1958-1984." c) It might be better to call the ice masses just glaciers instead of ice caps, since I assume the table includes not only those ice masses that are from a morphological point an ice cap? We agree and reformulated accordingly, thank you!

[revised manuscript text omitted]
 $\sim 100 \text{ m}$ to $440 \pm 80 \text{ m a.s.l.}$ (Figs. 3b and S3b), nearly coinciding with the
137 hypsometry peak (Fig. S3d). This rapidly expanded the ablation zone, exposing large areas to
138 increased melt. The subsequent firn line retreat strongly reduced the fraction of melt that refreezes
above the pre-1985 ELA (Fig. 3d), enhancing runoff 75% faster than melt ($+8.9 \text{ Gt yr}^{-1}$ vs $+6.7$
140 Gt yr^{-1}). Figure S4a shows the ELA change across Svalbard as a result of the post-1985 warming
($R = 0.82$; Fig. 4a). The ablation zone extent increases non-linearly with the upward migration
of the ELA (Fig. 4b), reflecting the proximity of the hypsometry peak (Figs. 3b, c). The size
of the ablation zone in turn governs meltwater production (Fig. 4c), since most of the melt is
produced over low-lying marginal glaciers exposing dark bare ice (Fig. S4b). In the absence of
refreezing, the low albedo of exposed ice increases melt through enhanced absorption of incoming
solar radiation, in turn driving the runoff increase. Most remarkably, increased melt triggers a
pronounced non-linear decrease in refreezing capacity (Fig. 4d), as i) the firn line retreat strongly
reduces the firn area hence limiting meltwater retention, and ii) meltwater fills the pore space of
the remaining firn through refreezing. These mechanisms could likely be reinforced by increased
rainfall episodes in a warmer climate, further reducing firn refreezing capacity³⁰.
Regionally, the upward migration of the ELA is largest in the northernmost sectors, e.g. NE (+130
152 m) and AF (+120 m), compared to southern sectors with an average of +85 m (Tables S2 and S3).
As a result, the ablation zone also grew fastest in the north, e.g. NE (+73%), VF (+91%) and
notably AF (+137%; Fig. S4a) compared to southern sectors (+48% on average; Tables S2 and
S3). For the northern sectors, this resulted in a 66% to 71% runoff increase after 1985, i.e. well
above the Svalbard average (+55%; Tables S2 and S3). These three northernmost sectors exhibit a
stronger response to atmospheric warming because of a pronounced decline in refreezing capacity
across their accumulation zones (-40% locally; Figs. 4d, e), increasing runoff at all elevations (Fig.
S4b). These results are in line with the study of Van Pelt et al. (2019) (see their Fig. 9d)²⁷. Since
it has the largest accumulation zone, the strongest sensitivity to atmospheric warming is found for
Austfonna ice cap (AF sector), containing a third ($\sim 2,500 \text{ km}^3$)¹⁶ of the total ice volume in the
archipelago. In contrast, for regions with smaller accumulation zones (NW and SS) or that had
already lost most of their refreezing capacity before 1985 (BE; Table S2), the runoff increase is
restricted to the margins (Fig. S4b), and primarily driven by ablation zone expansion rather than
loss of refreezing capacity (Fig. 4c).
The fact that the ELA now **fluctuates** around the hypsometry maximum makes Svalbard glaciers
highly sensitive to changes in atmospheric temperature. During warm summers, the ablation zone
now covers more than half of the surface area of most ice caps (Fig. 3c). In the warm summer of
2013, the ablation zone even covered 77% of the **land ice** area (Fig. 5b), almost **twice** the post-1985
average (44%; Table S3). This pronounced expansion stems from the fact that in 2013 the ELA
moved to 590 m a.s.l., i.e. **above** the hypsometry peak (Fig. S3d). Consequently, the refreezing
capacity dropped to 28% (2013), more than doubling runoff compared to previous years (47 Gt
173 yr^{-1} ; Fig. 3a). **We conclude that the post-1985 decline in refreezing capacity will persist under**
174 **continued warming:** a temporary return to pre-1985 SMB values in the period 2005-2012 (Figs. 3a
175 and 5a) did not lead to the recovery of the **refreezing capacity** (Fig. 3d). **At the current mass loss**
**rate ($19.4 \pm 3.4 \text{ Gt yr}^{-1}$ for 2013-2018), Svalbard glaciers would completely melt within the next**
**400 years.**
**Methods**
**Regional climate model and statistical downscaling.** We use the outputs of the Regional At-
mospheric Climate Model (RACMO2.3)²⁹ as input to the statistical downscaling procedure²⁸.
RACMO2.3 is run at 11 km spatial resolution for the period 1958-2018. The model incorporates
the dynamical core of the High-Resolution Limited Area Model (HIRLAM)³¹ and the physics
 of the European Centre for Medium-Range Weather Forecasts-Integrated Forecast (ECMWF-IFS
 cycle CY33r1)³². RACMO2.3 includes a multi-layer snow module simulating melt, water per-
 colation, retention and refreezing in firm, as well as runoff³³. The model accounts for dry snow
 densification³⁴, drifting snow erosion and sublimation³⁵, and explicitly simulates snow albedo
 ³⁶. **In this study we refer to 'surface mass balance' (SMB) as both the local (kg m²) and spatially**
 **integrated (Gt yr⁻¹) sum of:**

$$\text{SMB} = \text{PR} - \text{RU} - \text{SU} - \text{ER} \quad (1)$$

where PR represents total precipitation including snowfall (SF) and rainfall (RA), RU meltwater
 runoff, SU total sublimation and ER the erosion from drifting snow. Liquid water from rain and
 melt (ME) that is not retained or refrozen in firm (RF) contributes to runoff:

$$\text{RU} = \text{ME} + \text{RA} - \text{RF} \quad (2)$$

[revised manuscript text omitted]

edge base for climate adaptation. *Norwegian Environment Agency*
- (Miljødirektoratet), *NCSS report M-1242*, 208 pp. (2019). URL
www.miljodirektoratet.no/globalassets/publikasjoner/M741/M741.pdf.
- 4. Meredith, M. *et al.* *Polar Regions. In: IPCC Special Report on the Ocean and Cryosphere in*
*a Changing Climate* (IPCC SROCC In press., 2019).
- 5. Mèmin, A., Rogister, Y., Hinderer, J., Omang, O. C. & Luck, B. Global Glacier Mass Loss
During the GRACE Satellite Mission (2002-2016). *Geophysical Journal International* **184**,
293 1119 – 1130 (2011).
- 6. Jacob, T., Wahr, J., Pfeffer, W. T. & Swenson, S. Recent contributions of glaciers and ice caps
to sea level rise. *Nature* **514**, 514 – 518 (2012).
- 7. Gardner, A. S. *et al.* A reconciled estimate of glacier contributions to sea level rise: 2003 to
2009. *Science* **340**, 852 – 857 (2013).
- 8. Matsuo, K. & Heiki, K. Current Ice Loss in Small Glacier Systems of the Arctic Islands
(Iceland, Svalbard, and the Russian High Arctic) from Satellite Gravimetry. *Terrestrial, At-*
*mospheric and Oceanic sciences journal* **24**, 657 – 670 (2013).
- 9. Wouters, B., Chambers, D. & Schrama, E. J. O. GRACE observes small-scale mass loss in
Greenland. *Geophysical Research Letters* **35**, L20501 (2008).
- 10. Wouters, B., Gardner, A. & Moholdt, G. Global Glacier Mass Loss During the GRACE
Satellite Mission (2002-2016). *Frontiers in Earth Science* **7**, 11 p. (2019).
- 11. Zemp, M. *et al.* Global glacier mass changes and their contributions to sea-level rise from
1961 to 2016. *Nature* **568**, 382 – 386 (2019).
- 12. Schuler, T. V. *et al.* Reconciling Svalbard Glacier Mass Balance. *Frontiers in Earth Science*
**8**, 16 pp. (2020).
- 13. Blaszczyk, M., Jania, J. & Hagen, J. O. Tidewater glaciers of Svalbard: Recent changes and
estimates of calving fluxes. *Polish Polar Research* **30**, 85 – 142 (2009).
- 14. Hagen, J. O., Kohler, J., Melvold, K. & Winther, J.-G. Glaciers in Svalbard: mass balance,
runoff and freshwater flux. *Polar Research* **22**, 145 – 159 (2003).
- 15. Farnsworth, W. R., Ingólfsson, Ó., Retelle, M. & Schomacker, A. Over 400 previously undoc-
umented Svalbard surge-type glaciers identified. *Geomorphology* **264**, 52 – 60 (2016).
- 16. McMillan, M. *et al.* Rapid dynamic activation of a marine-based arctic ice cap. *Geophysical*
*Research Letters* **41**, 8902 – 8909 (2014).
- 17. Sund, M., Lauknes, T. R. & Eiken, T. Surge dynamics in the Nathorstbreen glacier system,
Svalbard. *The Cryosphere* **8**, 623 – 638 (2014).
- 18. Sevestre, H. *et al.* Tidewater Glacier Surges Initiated at the Terminus. *Journal of Geophysical*
*Research-Earth Surface* **123**, 1035 – 1051 (2018).
- 19. Nuth, C. *et al.* Dynamic vulnerability revealed in the collapse of an Arctic tidewater glacier.
*Scientific Reports* **9**, 13 pp. (2019).
- 20. Dunse, T. *et al.* Glacier-surge mechanisms promoted by a hydro-thermodynamic feedback to
summer melt. *The Cryosphere* **9**, 197 – 215 (2015).
- 21. Lang, C., Fettweis, X. & Erpicum, M. Stable climate and surface mass balance in Svalbard
over 1979–2013 despite the Arctic warming. *The Cryosphere* **9**, 83 – 101 (2015).
- 22. Aas, K. S. *et al.* The climatic mass balance of Svalbard glaciers: a 10-year simulation with a
coupled atmosphere–glacier mass balance model. *The Cryosphere* **10**, 1089 – 1104 (2016).
- 23. Noël, B. *et al.* A tipping point in refreezing accelerates mass loss of Greenland’s glaciers and
ice caps. *Nature Communications* **8**, 14730 (2017).
- 24. Noël, B. *et al.* Six decades of glacial mass loss in the Canadian Arctic Archipelago. *Journal*
*of Geophysical Research Earth Surface* **123**, 1430 – 1449 (2018).
- 25. Möller, M. & Kohler, J. Differing Climatic Mass Balance Evolution Across Svalbard Glacier
Regions Over 1900–2010. *Frontiers in Earth Science* **6**, 20 p. (2018).
- 26. Østby, T. I. *et al.* Diagnosing the decline in climatic mass balance of glaciers in Svalbard over
1957–2014. *The Cryosphere* **11** (2017).
- 27. Van Pelt, W. *et al.* A long-term dataset of climatic mass balance, snow conditions, and runoff
in Svalbard (1957–2018). *The Cryosphere* **13**, 2259 – 2280 (2019).
- 28. Noël, B. *et al.* A daily, 1 km resolution data set of downscaled Greenland ice sheet surface
mass balance (1958–2015). *The Cryosphere* **10**, 2361 – 2377 (2016).
- 29. Noël, B. *et al.* Evaluation of the updated regional climate model RACMO2.3: summer snow-
fall impact on the Greenland Ice Sheet. *The Cryosphere* **9**, 1831 – 1844 (2015).
- 30. Van Pelt, W. & Kohler, J. Modelling the long-term mass balance and firn evolution of glaciers
around Kongsfjorden, Svalbard. *Journal of Glaciology* **61**, 731 – 744 (2015).
- 31. Undèn, P. *et al.* HIRLAM-5. *Scientific Documentation* (2002). Technical Report.
- 32. ECMWF-IFS. Part IV : PHYSICAL PROCESSES (CY33R1). *Technical Report* (2008).
- 33. Ettema, J. *et al.* Climate of the Greenland ice sheet using a high-resolution climate model -
Part 1: Evaluation. *The Cryosphere* **4**, 511 – 527 (2010).
- 34. Ligtenberg, S. R. M., Munneke, P. K., Noël, B. & van den Broeke, M. R. Brief communication:
Improved simulation of the present-day Greenland firn layer (1960–2016). *The Cryosphere* **12**,
351 1643 – 1649 (2018).

- 35. Lenaerts, J. T. M., van den Broeke, M. R., Angelen, J. H., van Meijgaard, E. & Déry, S. J.
Drifting snow climate of the Greenland ice sheet: a study with a regional climate model. *The*
*Cryosphere* **6**, 891 – 899 (2012).
- 36. Van Angelen, J. H. *et al.* Sensitivity of Greenland Ice Sheet surface mass balance to surface
albedo parameterization: a study with a regional climate model. *The Cryosphere* **6**, 1175 –
1186 (2012).
- 37. Cogley, J. *et al.* *Glossary of Glacier Mass Balance and related terms* (IHP-VII Technical
Documents in Hydrology No. 86, IACS Contribution No. 2, UNESCO-IHP, Paris, 2011).
- 38. Uppala, S. M. *et al.* The ERA-40 re-analysis. *Quarterly Journal of the Royal Meteorological*
*Society* **131**, 2961 – 3012 (2005).
- 39. Dee, D. P. *et al.* The ERA-Interim reanalysis: configuration and performance of the data
assimilation system. *Quarterly Journal of the Royal Meteorological Society* **137**, 553 – 597
(2011).
- 40. Pfeffer, W. T. *et al.* The Randolph Glacier Inventory: a globally complete inventory of glaciers.
*Journal of Glaciology* **60**, 537 – 552 (2014).
- 41. Noël, B., van de Berg, W. J., Lhermitte, S. & van den Broeke, M. R. Rapid ablation zone
expansion amplifies north Greenland mass loss. *Science Advances* **5**, eaaw0123 (2019).
- 42. Hagen, J., Melvold, K., Eiken, T., Isaksson, E. & Lefauconnier, B. Mass balance methods on
Kongsvegen, Svalbard. *Geografiska Annaler: Series A, Physical Geography* **81**, 593 – 601
(1999).
- 43. Kohler, J. *et al.* Acceleration in thinning rate on western Svalbard glaciers. *Geophysical*
*Research Letters* **34**, L18502 (2007).
- 44. Grabiec, M., Jania, J. A., Puczko, D., Kolondra, L. & Budzik, T. Surface and bed morphology
of Hansbreen, a tidewater glacier in Spitsbergen. *Polish Polar Research* **33**, 111 – 138 (2012).
- 45. Van Pelt, W. J. J. *et al.* Dynamic Response of a High Arctic Glacier to Melt and Runoff
Variations. *Geophysical Research Letters* **45**, 4917 – 4926 (2018).
- 46. Wouters, B. *et al.* Dynamic thinning of glaciers on the Southern Antarctic Peninsula. *Science*
**348**, 899 – 903 (2015).
**Acknowledgements** B. Noël was funded by NWO VENI grant VI.Veni.192.019. C. L. Jakobs, C. H.
Reijmer, W. J. van de Berg, and M. R. van den Broeke acknowledge support from the Polar Programme
of the Netherlands Organization for Scientific Research (NWO/ALW) and the Netherlands Earth System
Science Centre (NESSC). B. Wouters was funded by NWO VIDI grant 016.Vidi.171.063.
**Data availability** Daily downscaled SMB and components at 500 m spatial resolution (1958-2018) and
other data required to reproduce the tables and figures presented in the manuscript are available from the
authors upon request and without conditions. SMB components include total precipitation (snowfall and
rainfall), snowfall, runoff, melt, refreezing and retention, total sublimation and snow drift erosion.
**Code availability** RACMO2.3 is presented in Ref. ²⁹ and the statistical downscaling technique is de-
scribed in Ref. ²⁸.
**Authors contribution** B.N. prepared the manuscript, carried out the RACMO2.3 simulation and produced
the downscaled dataset at 500 m. C.L.J. helped conducting and analysing the RACMO2.3 simulations. B.N.,
392 W.J.B. and M.R.B. conceived the downscaling procedure and analysed the data. W.J.J.P., J.K., J.O.H., B.L.
and C.H.R. provided the Svalbard in situ SMB dataset and the S0 Terreng DEM. S.L. processed the 500
394 m MODIS albedo product. B.W. produced and analysed the GRACE, ICESat and CryoSat-2 datasets. All
395 authors commented on the manuscript.

**Competing Interests** The authors declare that they have no competing financial interests.
**Correspondence** Correspondence and requests for materials should be addressed to Brice Noël. (email:
b.p.y.noel@uu.nl).

Figure 1: **Svalbard surface mass balance and hypsometry.** (a) **Modelled surface mass balance (SMB) statistically downscaled to 500 m spatial resolution, averaged for the period 1958-2018.** **Orange dots** locate the 101 stakes used for model evaluation (Fig. S2a). The sectors of Svalbard evaluated in Fig. S2b are also outlined. (b) Hypsometry of six Arctic **ice masses**: Svalbard (S0 Terreng DEM), Iceland (Arctic DEM), North and South Canadian Arctic Archipelago (Canadian DEM)²⁴, Greenland ice sheet (GIMP DEM)⁴¹, Greenland peripheral glaciers and ice caps (GIMP DEM)²³. The x-axis shows the **glacier area in each 100 m elevation band** as a fraction of the total ice area of that region (%).

Figure 2: **Cumulative mass change of Svalbard glaciers and contribution to sea level rise.** Time series of monthly cumulative **modelled SMB**, **measured cumulative solid ice discharge (D)**^{11,12} and **reconstructed cumulative mass balance (MB = SMB minus D)** for the period 1958-2018. **Observed mass change** derived from GRACE (2002-2016), ICESat (2003-2009) and CryoSat-2 (2010-2018) are also shown. For clarity, GRACE data are shown with a positive offset of 100 Gt. The right y-axis translates **Svalbard cumulative mass balance** into global sea level rise equivalent. Figure S2c zooms in on the satellite period (2003-2018).

Figure 3: **Ablation zone expansion and reduced refreezing capacity.** (a) Time series of annual SMB and components including surface melt, runoff, **total precipitation** and refreezing for the period 1958-2018. (b) Time series of annual ELA for the whole of Svalbard (black) and individual sectors (Fig. 1a, orange band). (c) Time series showing the modelled ablation zone area, the modelled and observed (MODIS) bare ice area as a fraction of the total Svalbard land ice area (%). (d) Time series of annual refreezing capacity for the whole of Svalbard (black) and individual sectors (cyan band). Dashed lines show averages for the periods 1958-1984 and 1985-2018. The grey shade highlights the period 2005-2012 when Svalbard SMB temporarily returned to the pre-1985 SMB conditions. Dashed grey lines represent the 2005-2012 mean conditions.

Figure 4: **Sensitivity of Svalbard refreezing capacity to atmospheric warming.** Scatterplots showing Svalbard-wide correlations between (a) **June-July-August 2 m air temperature anomaly (1985-2018 minus 1958-1984)** and ELA; (b) ELA and ablation zone area; (c) ablation zone area and surface melt, and (d) melt and firm refreezing capacity. Statistics include number of records (N), correlation (R) and fitting parameters (a,b,c). (e) Post-1985 change in refreezing capacity (%; 1985-2018 minus 1958-1984). ELA for the period 1985-2018 is also shown as a black line.

Figure 5: **Ablation zone expansion in summer 2013.** (a) SMB average for the period 2005-2012, with SMB conditions similar to 1958-1984. (b) SMB for year 2013 highlighting how fast the ablation zone expands when the ELA migrates well above the hypsometry maximum (~450 m a.s.l.). From the thickest to the thinnest, black lines outline the ELA for periods 1958-1984, 1985-2018 (a and b) and year 2013 (b only).

Supplementary Information: "Low elevation of Svalbard glaciers drives high mass loss variability"

Brice Noël*¹, C. L. Jakobs¹, W. J. J. van Pelt², S. Lhermitte³, B. Wouters^{1,3}, J. Kohler⁴, J. O. Hagen⁵, B. Luks⁶, C. H. Reijmer¹, W. J. van de Berg¹, & M. R. van den Broeke¹

¹*Institute for Marine and Atmospheric research Utrecht, Utrecht University, 3584 CC Utrecht, Netherlands.*

²*Department of Earth Sciences, Uppsala University, SE 75236 Uppsala, Sweden.*

³*Department of Geoscience & Remote Sensing, Delft University of Technology, 2600 AA Delft, Netherlands.*

⁴*Norwegian Polar Institute, N-9296 Tromsø, Norway.*

⁵*Department of Geosciences, University of Oslo, 0371 Oslo, Norway.*

⁶*Institute of Geophysics, Polish Academy of Sciences, 01-452 Warsaw, Poland.*

Supplementary Figure 1: **Topography of the Svalbard archipelago.** Surface elevation (m a.s.l.) derived from the S0 TerrenG DEM of Svalbard at 20 m spatial resolution (Norwegian Polar Institute) and down-sampled to a 500 m grid.

Supplementary Figure 2: **Model evaluation using in situ and remote sensing measurements.** (a) Comparison between modelled and observed SMB at 101 stakes (Fig. 1a). The red dashed line represents the regression including all measurements. (b) Comparison between modelled and observed bare ice **area** for individual sectors. The grey dashed line corresponds to the regression using all measurements. (c) Time series of monthly cumulative **reconstructed** mass balance (MB = SMB minus solid ice discharge) overlapping the satellite period (2003-2018): GRACE (2003-2016), ICESat (2003-2009) and CryoSat-2 (2010-2018). The inset in Fig. S2c shows the comparison between modelled and remotely sensed monthly cumulative mass change from GRACE, ICESat and CryoSat-2. Regressions are shown as dashed red (GRACE) and green (ICESat/CryoSat-2) line. Statistics including the number of observations (N), slope (b0) and intercept (b1) of the regression line, coefficient of determination (R^2), RMSE and mean bias between model and observations are also listed.

Supplementary Figure 3: **Upward migration of the firn line.** Vertical profile of integrated SMB and components including **total precipitation (snowfall and rainfall)**, rainfall, melt, runoff and re-freezing for the periods (a) 1958-1984, (b) 1985-2018, and (c) the difference between the two periods (1985-2018 minus 1958-1984). (d) Hypsometry of Svalbard ice caps, i.e. integrated ice-covered area within 100 m elevation bins. The grey band spans the minimum and maximum ELA (SMB = 0) of individual sectors for the periods 1958-1984 and 1985-2018.

Supplementary Figure 4: **Ablation zone expansion and runoff change.** (a) SMB of Svalbard ice caps averaged for the period 1985-2018. The thick and thin black lines outline the ELA (local SMB = 0) for periods 1958-1984 and 1985-2018. (b) Post-1985 change in meltwater runoff (1985-2018 minus 1958-1984). The black line outlines the 1985-2018 ELA.

Supplementary Table 1: **Model evaluation using recent mass change estimates.** Comparison between mass balance (MB = SMB minus D) from the current study and previous **geodetic, GRACE, model and observation-based mass change estimates.** Models include the **Weather Research and Forecasting model (WRF), the Modèle Atmosphérique Régional (MAR), a Positive Degree Day (PDD) and two Energy Balance Models (EBM).** In our study, solid ice discharge (D) is derived from Ref. ¹ before 2012 and combined Refs. ^{1,2} afterwards.

References	Method	Period	Units	Estimate	This study
Moholdt et al. (2010) ³	Geodetic	2003-2008	Gt yr ⁻¹	-4.1 ± 1.4	-7.1 ± 3.4
Zemp et al. (2019) ⁴	Geodetic	2006-2016	Gt yr ⁻¹	-16.0 ± 8.0	-9.7 ± 3.4
Wouters et al. (2008) ⁵	GRACE	2003-2008	Gt yr ⁻¹	-8.8 ± 3.0	-7.1 ± 3.4
Mèmin et al. (2011) ⁶	GRACE	2003-2009	Gt yr ⁻¹	-9.1 ± 1.0	-6.0 ± 3.4
Gardner et al. (2013) ⁷	GRACE	2003-2009	Gt yr ⁻¹	-6.8 ± 2.0	-6.0 ± 3.4
Jacob et al. (2012) ⁸	GRACE	2003-2010	Gt yr ⁻¹	-3.0 ± 2.0	-5.7 ± 3.4
Matsuo et al. (2013) ⁹	GRACE	2004-2008	Gt yr ⁻¹	-6.8 ± 3.7	-3.6 ± 3.4
Matsuo et al. (2013) ⁹	GRACE	2004-2012	Gt yr ⁻¹	-3.7 ± 3.0	-4.7 ± 3.4
Wouters et al. (2019) ¹⁰	GRACE	2002-2016	Gt yr ⁻¹	-7.2 ± 1.4	-9.3 ± 3.4
Aas et al. (2016) ¹¹	WRF	2003-2013	Gt yr ⁻¹	-8.7	-9.1 ± 3.4
Lang et al. (2015) ¹²	MAR	1979-2013	Gt yr ⁻¹	-8.4	-8.0 ± 3.4
Möller et al. (2018) ¹³	PDD	1957-2010	Gt yr ⁻¹	1.0	-4.1 ± 3.4
Østby et al. (2017) ¹⁴	EBM	1957-2014	Gt yr ⁻¹	-4.0	-5.0 ± 3.4
Van Pelt et al. (2019) ¹⁵	EBM	1957-2018	Gt yr ⁻¹	-3.0	-5.9 ± 3.4
Schuler et al. (2020) ¹⁶	Data upscaling	2000-2019	Gt yr ⁻¹	-8.0 ± 6	-11.4 ± 3.4

Supplementary Table 2: **State of Svalbard glaciers pre-1985**. This table lists the mass balance (MB = SMB minus D), SMB and components, firm refreezing capacity, ablation zone area (i.e. as a fraction of the total glacier area), and ELA (SMB = 0) for individual sectors and the whole of Svalbard averaged over the period 1958-1984. Solid ice discharge (D) is estimated from Ref. ¹. **ELA uncertainty is estimated as one standard deviation of the period 1958-1984**.

1958-1984	Units	NW	NE	VF	AF	BE	SS	Svalbard
MB	Gt yr ⁻¹	-	-	-	-	-	-	-0.4 ± 3.4
SMB	Gt yr ⁻¹	0.3 ± 0.3	2.0 ± 0.4	0.5 ± 0.1	2.9 ± 0.3	0.2 ± 0.1	0.6 ± 0.3	6.3 ± 1.6
Precipitation	Gt yr ⁻¹	4.2	5.5	1.4	5.1	1.6	4.8	23.0
Runoff	Gt yr ⁻¹	3.9	3.5	0.9	2.1	1.5	4.2	16.3
Melt	Gt yr ⁻¹	7.4	6.8	1.5	4.3	1.9	6.4	28.7
Refreezing	Gt yr ⁻¹	4.5	4.1	0.8	2.7	0.7	3.4	16.5
Refreezing capacity	%	57.6	58.7	52.1	60.3	37.2	48.9	54.4
Ablation zone area	%	35.4	25.1	21.5	11.6	36.0	34.8	27.1
ELA	m	470 ± 65	380 ± 100	270 ± 84	220 ± 60	280 ± 80	340 ± 50	350 ± 60

Supplementary Table 3: **State of Svalbard glaciers post-1985**. This table lists the mass balance (MB = SMB minus D), SMB and components, firm refreezing capacity, ablation zone area (i.e. as a fraction of the total glacier area), and ELA (SMB = 0) for individual sectors and the whole of Svalbard averaged over the period 1985-2018. Solid ice discharge (D) is estimated from Ref. ¹ before 2012 and combined Refs. ^{1,2} afterwards. **ELA uncertainty is estimated as one standard deviation of the period 1985-2018**.

1985-2018	Units	NW	NE	VF	AF	BE	SS	Svalbard
MB	Gt yr ⁻¹	-	-	-	-	-	-	-10.2 ± 3.4
SMB	Gt yr ⁻¹	-1.6 ± 0.3	-0.2 ± 0.4	0.0 ± 0.1	1.4 ± 0.3	-0.6 ± 0.1	-1.3 ± 0.3	-2.6 ± 1.6
Precipitation	Gt yr ⁻¹	4.2	5.6	1.3	5.0	1.5	4.7	22.8
Runoff	Gt yr ⁻¹	5.9	5.8	1.3	3.6	2.2	6.0	25.2
Melt	Gt yr ⁻¹	8.9	8.6	1.8	5.5	2.5	7.6	35.4
Refreezing	Gt yr ⁻¹	4.2	3.9	0.7	2.5	0.6	3.0	15.1
Refreezing capacity	%	45.0	43.3	39.2	44.8	23.9	36.1	40.6
Ablation zone area	%	49.0	43.3	41.0	27.5	61.0	51.5	43.9
ELA	m	550 ± 65	510 ± 130	360 ± 100	340 ± 110	370 ± 85	420 ± 55	440 ± 80

Supplementary References

1. Blaszczyk, M., Jania, J. & Hagen, J. O. Tidewater glaciers of Svalbard: Recent changes and estimates of calving fluxes. *Polish Polar Research* **30**, 85 – 142 (2009).
2. Dunse, T. *et al.* Glacier-surge mechanisms promoted by a hydro-thermodynamic feedback to summer melt. *The Cryosphere* **9**, 197 – 215 (2015).
3. Moholdt, G., Nuth, C., Hagen, J. O. & Kohler, J. Recent elevation changes of Svalbard glaciers derived from ICESat laser altimetry. *Remote Sensing of Environment* **114**, 2756 – 2767 (2010).
4. Zemp, M. *et al.* Global glacier mass changes and their contributions to sea-level rise from 1961 to 2016. *Nature* **568**, 382 – 386 (2019).
5. Wouters, B., Chambers, D. & Schrama, E. J. O. GRACE observes small-scale mass loss in Greenland. *Geophysical Research Letters* **35**, L20501 (2008).
6. Mèmin, A., Rogister, Y., Hinderer, J., Omang, O. C. & Luck, B. Global Glacier Mass Loss During the GRACE Satellite Mission (2002-2016). *Geophysical Journal International* **184**, 1119 – 1130 (2011).
7. Gardner, A. S. *et al.* A reconciled estimate of glacier contributions to sea level rise: 2003 to 2009. *Science* **340**, 852 – 857 (2013).
8. Jacob, T., Wahr, J., Pfeffer, W. T. & Swenson, S. Recent contributions of glaciers and ice caps to sea level rise. *Nature* **514**, 514 – 518 (2012).
9. Matsuo, K. & Heiki, K. Current Ice Loss in Small Glacier Systems of the Arctic Islands (Iceland, Svalbard, and the Russian High Arctic) from Satellite Gravimetry. *Terrestrial, Atmospheric and Oceanic sciences journal* **24**, 657 – 670 (2013).
10. Wouters, B., Gardner, A. & Moholdt, G. Global Glacier Mass Loss During the GRACE Satellite Mission (2002-2016). *Frontiers in Earth Science* **7**, 11 p. (2019).
11. Aas, K. S. *et al.* The climatic mass balance of Svalbard glaciers: a 10-year simulation with a coupled atmosphere–glacier mass balance model. *The Cryosphere* **10**, 1089 – 1104 (2016).
12. Lang, C., Fettweis, X. & Erpicum, M. Stable climate and surface mass balance in Svalbard over 1979–2013 despite the Arctic warming. *The Cryosphere* **9**, 83 – 101 (2015).
13. Möller, M. & Kohler, J. Differing Climatic Mass Balance Evolution Across Svalbard Glacier Regions Over 1900–2010. *Frontiers in Earth Science* **6**, 20 p. (2018).
14. Østby, T. I. *et al.* Diagnosing the decline in climatic mass balance of glaciers in Svalbard over 1957–2014. *The Cryosphere* **11** (2017).
15. Van Pelt, W. *et al.* A long-term dataset of climatic mass balance, snow conditions, and runoff in Svalbard (1957–2018). *The Cryosphere* **13**, 2259 – 2280 (2019).

- 35 16. Schuler, T. V. *et al.* Reconciling Svalbard Glacier Mass Balance. *Frontiers in Earth Science*
36 **8**, 16 pp. (2020).